# Threshold-aware Learning to Generate Feasible Solutions for Mixed Integer Programs

## Abstract

Finding a high-quality feasible solution to a combinatorial optimization (CO) problem in a limited time is challenging due to its discrete nature. Recently, there has been an increasing number of machine learning (ML) methods for addressing CO problems. Neural diving (ND) is one of the learning-based approaches to generating partial discrete variable assignments in Mixed Integer Programs (MIP), a framework for modeling CO problems. However, a major drawback of ND is a large discrepancy between the ML and MIP objectives, *i.e.*, misalignment between the variable value classification accuracy and primal bound. Our study investigates that a specific range of variable assignment rates (*coverage*) yields high-quality feasible solutions, where we suggest optimizing the coverage bridges the gap between the learning and MIP objectives. Consequently, we introduce a post-hoc method and a learning-based approach for optimizing the coverage. A key idea of our approach is to jointly learn to restrict the coverage search space and to predict the coverage in the learned search space. Experimental results demonstrate that learning a deep neural network to estimate the coverage for finding high-quality feasible solutions achieves state-of-the-art performance in NeurIPS ML4CO datasets. In particular, our method shows outstanding performance in the workload apportionment dataset, achieving the optimality gap of 0.45%, a ten-fold improvement over SCIP within the one-minute time limit.

## 1 Introduction

Mixed integer programming (MIP) is a mathematical optimization model to solve combinatorial optimization (CO) problems in diverse areas, including transportation (Applegate et al., 2003), finance (Mansini et al., 2015), communication (Das et al., 2003), manufacturing (Pochet & Wolsey, 2006), retail (Sawik, 2011), and system design (Maulik et al., 1995). A workhorse of the MIP solver is a group of heuristics involving primal heuristics (Berthold, 2006; Fischetti et al., 2005) and branching heuristics (Achterberg et al., 2005; 2012). Recently, there has been a growing interest in applying data-driven methods to complement heuristic components in the MIP solvers: node selection (He et al., 2014), cutting plane (Tang et al., 2020), branching (Khalil et al., 2017; Balcan et al., 2018; Gasse et al., 2019), and primal heuristics (Nair et al., 2020; Xavier et al., 2021). In particular, finding a high-quality feasible solution in a shorter time is an essential yet challenging task due to its discrete nature. Nair et al. (2020) suggests Neural diving (ND) with a variant of SelectiveNet (Geifman & El-Yaniv, 2019) to jointly learn diving style primal heuristics to generate feasible solutions and adjust a discrete variable selection rate for assignment, referred to as *coverage*. ND partially assigns the discrete variable values in the original MIP problem and delegates the remaining sub-MIP problem to the MIP solver. However, there are two problems in ND. First, the ND model that shows higher variable value classification accuracy for the collected MIP solution, defined as the ratio of the number of correctly predicted variable values to the total number of predicted variable values, does not necessarily generate a higher-quality feasible MIP solution. We refer to this problem as discrepancy between ML and MIP objectives. Secondly, obtaining the appropriate ND coverage that yields a high quality feasible solution necessitates the training of multiple models with varying target coverages, a process that is computationally inefficient.

In this work, we suggest that the coverage is a surrogate measure to bridge the gap between the ML and MIP objectives. A key observation is that the MIP solution obtained from the ML model exhibits a sudden shift in both solution feasibility and quality concerning the coverage. Building on our insight, we propose a simple variable selection strategy called Confidence Filter (CF) to overcome SelectiveNet's training inefficiency problem. While primal heuristics rely on LP-relaxation solution values, CF fixes the variables whose corresponding neural network output satisfies a specific cutoff value. It follows that CF markedly reduces the total ND training cost since it only necessitates a single ND model without SelectiveNet. At the same time, determining the ideal CF cutoff value with regard to the MIP solution quality can be achieved in linear time in the worst-case scenario concerning the number of discrete variables, given that the MIP model assessment requires a constant time. To relieve the CF evaluation cost, we devise a learning-based method called Threshold-aware Learning (TaL) based on threshold function theory (Bollobás, 1981). In TaL, we propose a new loss function to estimate the coverage search space in which we optimize the coverage. TaL outperforms other methods in various MIP datasets, including NeurIPS Machine Learning for Combinatorial Optimization (ML4CO) datasets Gasse et al. (2022).

Our contributions are as follows.

1) To the best of our knowledge, our work is the first attempt to investigate the role of threshold functions in connection with the importance of coverage optimization in learning to generate MIP solutions.

2) We devise a new ML framework for data-driven coverage optimization based on threshold function theory. Our method effectively relieves the gap between the ML and MIP objectives by utilizing the coverage as a surrogate measure.

3) We empirically demonstrate that our method outperforms the existing methods in various MIP datasets. In the maritime inventory routing dataset, our method achieves an optimality gap that is roughly twice as improved as that attained by ND.

## 2 Preliminaries

### 2.1 Notations on Mixed Integer Programming

Let $n$ be the number of variables, $m$ be the number of linear constraints, $\mathbf{x} = [x_1, \ldots, x_n] \in \mathbb{R}^n$ be the vector of variable values, $\mathbf{c} \in \mathbb{R}^n$ be the vector of objective coefficients, $\mathbf{A} \in \mathbb{R}^{m \times n}$ be the linear constraint coefficient matrix, $\mathbf{b} \in \mathbb{R}^m$ be the vector of linear constraint upper bound. Let $\{\mathbf{e}_i = \mathbf{e}_i^{(n)} \in \mathbb{R}^n : i = 1, \ldots, n\}$ be the Euclidean unit basis vector of $\mathbb{R}^n$ and $[n] := \{1, 2, \ldots, n\}$. Here we can decompose the variable vector as $\mathbf{x} = \sum_{i \in B} x_i \mathbf{e}_i$.

A Mixed Integer Program (MIP) is a mathematical optimization problem in which the variables are constrained by linear and integrality constraints. Let $r$ be the number of discrete variables. We express an MIP problem $M = (\mathbf{A}, \mathbf{b}, \mathbf{c})$ as follows:

$$\min_{\mathbf{x}} \mathbf{c}^\top \mathbf{x} \tag{1}$$
$$\text{subject to } \mathbf{A}\mathbf{x} \leq \mathbf{b}$$

where $\mathbf{x} = [x_1, \ldots, x_n] \in \mathbb{Z}^r \times \mathbb{R}^{n-r}$. We say $\mathbf{x} \in \mathbb{Z}^r \times \mathbb{R}^{n-r}$ is a *feasible solution* and write $\mathbf{x} \in \mathcal{R}(M)$ if the linear and integrality constraints hold. We call $\mathbf{x} \in \mathbb{R}^n$ an *LP-relaxation solution* and write $\mathbf{x} \in \bar{\mathcal{R}}(M)$ when $\mathbf{x}$ satisfies the linear constraints regardless of the integrality constraints.

### 2.2 Neural diving

Neural diving (ND) (Nair et al., 2020) is a method to learn a generative model that emulates diving-style heuristics to find a high-quality feasible solution. ND learns to generate a Bernoulli distribution of a solution's integer variable values in a supervised manner. For simplicity of notation, we mean integer

variables' solution values when we refer to $\mathbf{x}$. The conditional distribution over the solution $\mathbf{x}$ given an MIP instance $M = (\mathbf{A}, \mathbf{b}, \mathbf{c})$ is defined as

$$p(\mathbf{x}|M) = \frac{\exp(-E(\mathbf{x};M))}{Z(M)}, \tag{2}$$

where $Z(M)$ is the partition function for normalization defined as

$$Z(M) = \sum_{x'} \exp\left(-E\left(x';M\right)\right), \tag{3}$$

and $E(\mathbf{x}; M)$ is the energy function over the solution $\mathbf{x}$ defined as

$$E(\mathbf{x};M) = \begin{cases} \mathbf{c}^\top \mathbf{x} & \text{if } \mathbf{x} \text{ is feasible} \\ \infty & \text{otherwise} \end{cases} \tag{4}$$

ND aims to model $p(\mathbf{x}|M)$ with a generative neural network $p_\theta(\mathbf{x}|M)$ parameterized by $\theta$. Assuming conditional independence between variables, ND defines the conditionally independent distribution over the solution as

$$p_\theta(\mathbf{x}|M) = \prod_{d \in \mathcal{I}} p_\theta\left(x_d|M\right), \tag{5}$$

where $\mathcal{I}$ is the set of dimensions of $\mathbf{x}$ that belongs to discrete variables and $x_d$ denotes the $d$-th dimension of $\mathbf{x}$. Given that the complete ND output may not always conform to the constraints of the original problem, ND assigns a subset of the predicted variable values and delegates the off-the-shelf MIP solver to finalize the solution. In this context, Nair et al. (2020) employs SelectiveNet (Geifman & El-Yaniv, 2019) to regularize the coverage for the discrete variable assignments in an integrated manner. Here, the coverage is defined as the number of integer variables that are predicted divided by the number of all integer variables. Regularized by the predefined coverage, the ND model jointly learns to select variables to assign and predict variable values for the selected variables.

## 3 Coverage Optimization

**Confidence Filter**    We propose Confidence Filter (CF) to control the coverage in place of the SelectiveNet in ND. CF yields a substantial enhancement in the efficiency of model training, since it only necessitates a single ND model and eliminates the need for SelectiveNet. In CF, we adjust the confidence score cutoff value of the ND model output $p_\theta(\mathbf{x}|M)$, where the confidence score is defined as $\max(p_\theta(x_d|M), 1 - p_\theta(x_d|M))$ for the corresponding variable $x_d$. Suppose the target cutoff value is $\Gamma$, and the coverage is $\rho$. Let $s_d$ represent the selection of the $d$-th variable. The value $x_d$ is selected to be assigned if $s_d = 1$. It follows that $\rho \propto 1/\Gamma$.

$$s_d = \begin{cases} 1 & \text{if } \max(p_\theta(x_d|M), 1 - p_\theta(x_d|M)) \geq \Gamma \\ 0 & \text{otherwise} \end{cases} \tag{6}$$

Our hypothesis is that a higher confidence score associated with each variable corresponds to a higher level of certainty in predicting the value for that variable with regard to the solution feasiblility and quality.

As depicted in Figure 1, the feasibility and quality of the solution undergo significant changes depending on the cutoff value of the confidence score. Figure 1a illustrates that a majority of the partial variable assignments attain feasibility when the confidence score cutoff value falls within the range from 0.95 to 1. As depicted in Figure 1b, the solutions obtained from the CF partial variable assignments outperform the default MIP solver (SCIP) for cutoff values within the range from 0.9 to 0.98. Therefore, identifying the appropriate cutoff value is crucial for achieving a high-quality feasible solution within a time limit.

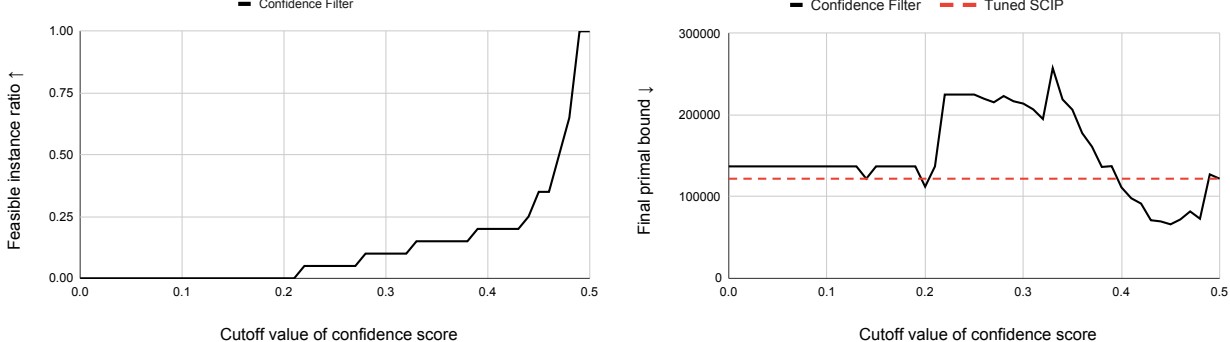

(a) Feasible instance ratio over the confidence score cutoff value. (b) Final primal bound over the confidence score cutoff value.

Figure 1: The feasible instance ratio and the final primal bound over the cutoff value of the Confidence Filter method at the 30-minute time limit. The feasible instance ratio is the number of test instances whose model-generated partial variable assignments are feasible, divided by the total number of test instances; the higher, the better. The final primal bound represents the solution quality; the lower, the better. In both (a) and (b), the feasibility and the solution quality dramatically change at a certain cutoff point. The dotted line in (b) represents the default MIP solver (SCIP) performance without Neural diving. In (b), the Confidence Filter outperforms SCIP at the confidence score cutoff value from around 0.4 to 0.48. The target dataset is Maritime Inventory Routing problems from (Papageorgiou et al., 2014; Gasse et al., 2022).

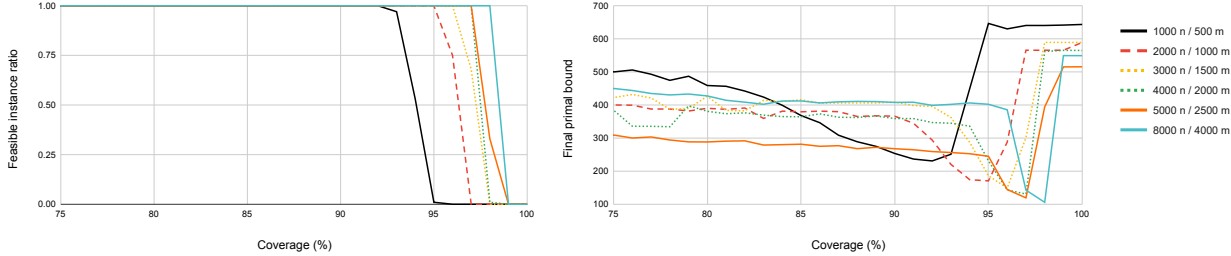

(a) Feasible instance ratio over the coverage. (b) Final primal bound over the coverage rate.

Figure 2: The threshold behavior of partial variable assignments with respect to coverage is examined in terms of the feasibility and final primal bound in set covering instances. We scale up the number of discrete variables $n$, and the number of constraints $m$, by $1\times$, $2\times$, $3\times$, $4\times$, $5\times$, and $8\times$. Each line in the plot corresponds to each problem size. We use the same model (ND without SelectiveNet) trained on the dataset of the smallest scale, $n = 1000, m = 500$. Both (a) and (b) exhibit consistent patterns with respect to the problem size, concerning the coverage where the threshold behavior occurs.

**Threshold coverage** Figure 2 shows an abrupt shift in the solution feasibility and quality within a specific range of coverage. Figure 2a and 2b demonstrate that the threshold coverage, representing the coverage at which a sudden shift in feasibility and quality occurs, follows a consistent pattern as the test problem sizes increase. Given the solution generated by the neural network, Figure 2b demonstrates that the optimal coverage defined as $\rho^* = \arg\min \text{PrimalBound}(\rho)$ is bounded by the feasibility threshold coverage in Figure 2a. We leverage the consistent pattern in the threshold coverage to estimate the optimal coverage.

### 3.1 Threshold-aware Learning

**Overview** The main objective of Threshold-aware Learning (TaL) is to learn to predict the optimal coverage $\rho^*$ that results in a high-quality feasible solution. We leverage the off-the-shelf MIP solver in TaL to use

the actual MIP objectives as a criterion to decide the optimal coverage. However, optimizing the coverage for large-scale MIP problems is costly since we need to find a feasible solution for each coverage rate at each training step. Therefore, we improve the search complexity for estimating $\rho^*$ by restricting the coverage search space that leads to high-quality feasible solutions. In TaL, we jointly learn to predict the optimal coverage and to restrict the coverage search space based on the feasibility and the quality of the partial variable assignments induced by the coverage. We demonstrate that the coverage is an effective surrogate measure to reduce discrepancies between the ML and MIP objectives. We also show that TaL reduces the CF's evaluation cost from linear to constant time.

**Random subset** We introduce a *random subset* $S(n, \rho)$ as a set-valued random variable. Here $n$ represents the index set $[n]$. For each element in $[n]$, $\rho = \rho(n) \in [0, 1]$ is the probability that the element is present. The selection of each element is statistically independent of the selection of all other elements. We call $\rho$ the coverage of the subset. Formally, the sample space $\Omega$ for $S(n, \rho)$ is a power set of $[n]$, and the probability measure for each subset $B \in \Omega$ assigns probability

$$\mathbb{P}(B) = \rho^l (1 - \rho)^{n-l}, \tag{7}$$

where $l$ is the number of elements in $B$. Given an MIP solution $\mathbf{x} \in \mathbb{R}^n$, we refer to $S(n, \rho)$ as a realization of the random variable that assigns the variable values of the selected dimensions of $\mathbf{x}$.

**Learning coverage search space** We approximate the optimal coverage $\rho^*$ with a graph neural network output $\hat{\rho}_\pi$. Also, $\hat{\rho}_\psi$ is a graph neural network output that models the threshold coverage at which the variable assignments of $\mathbf{x}$ switch from a feasible to an infeasible state. Similarly, $\hat{\rho}_\phi$ is a graph neural network output that estimates the threshold coverage for computing a criterion value of an LP-relaxation solution objective, which can be regarded as a lower bound of the solution quality. The LP-relaxation solution objective criterion pertains to a specific condition where the sub-MIP resulting from partial variable assignments is feasible and its LP-relaxation solution objective meets certain value. For formal definitions, see Definition A.3 and A.4.

$$\hat{\rho}_\psi, \hat{\rho}_\phi, \hat{\rho}_\pi = \texttt{GraphNeuralNet}(M) \tag{8}$$

Note that the neural networks $\hat{\rho}_\psi$, $\hat{\rho}_\phi$, and $\hat{\rho}_\pi$, share the same ND model weights. Let $\hat{\rho}_\Psi^{\mathrm{prob}}$ be a graph neural network output that predicts the probability of the partial variable assignments induced by the coverage $\hat{\rho}_\psi$ being feasible. Similarly, $\hat{\rho}_\Phi^{\mathrm{prob}}$ is a graph neural network output that predicts the probability of the partial variable assignments induced by the coverage $\hat{\rho}_\pi$ satisfying the LP-relaxation solution objective criterion. $\hat{\rho}_\Psi^{\mathrm{prob}}$ and $\hat{\rho}_\Phi^{\mathrm{prob}}$ share the same weights of the ND model.

$$\hat{\rho}_\Psi^{\mathrm{prob}}, \hat{\rho}_\Phi^{\mathrm{prob}} = \texttt{GraphNeuralNet}(M) \tag{9}$$

We define the indicator random variable that represents the state of the variable assignments.

$$I_{S(n,\rho)}^{\mathrm{feas}} = \begin{cases} 1, & \text{if } S(n,\rho) \text{ is feasible} \\ 0, & \text{otherwise} \end{cases} \quad I_{S(n,\rho)}^{\mathrm{LP\text{-}sat}} = \begin{cases} 1, & \text{if } S(n,\rho) \text{ satisfies the LP relaxation} \\ & \text{solution objective criterion} \\ 0, & \text{otherwise} \end{cases} \tag{10}$$

**Loss function** We propose a loss function to find the optimal coverage more efficiently. Let $\mathrm{BCE}(a, b) = -(a \log b + (1 - a) \log(1 - b))$ for $a, b \in (0, 1)$ be the binary cross entropy function. We implement the loss function using the negative log-likelihood function, where the minimization of the negative log-likelihood is the same as the minimization of binary cross entropy function. Let $t$ be the iteration step in the optimization loop. Let $\mathbf{x}(\tau, \rho)$ be the primal solution at time $\tau$ with initial variable assignments of $\mathbf{x}$ with coverage $\rho$. In practice, we obtain the optimal coverage as $\rho^* = \arg\min_{\rho \in [\hat{\rho}_\phi, \hat{\rho}_\psi]} \mathbf{c}^\top \mathbf{x}(\tau, \rho)$. To approximate the optimal coverage $\rho^*$ using a GNN model $\hat{\rho}_\pi$ parameterized by $\pi$, we learn the model parameters $\pi$ by minimizing the loss function $\mathcal{L}_{\mathrm{coverage}}(\pi)$ defined as

$$\mathcal{L}_{\mathrm{coverage}}(\pi) = \|\hat{\rho}_\pi - \rho^*\|_2^2 \tag{11}$$

Since the search cost for the optimal coverage is expensive, we restrict the coverage search space by approximating the threshold coverages induced by the feasibility event $I^{\text{feas}}_{S(n,\hat{\rho}_\psi)}$ and the LP-relaxation solution objective satisfiability event $I^{\text{LP-sat}}_{S(n,\hat{\rho}_\pi)}$. To approximate the threshold coverages using GNN models $\hat{\rho}_\psi$ and $\hat{\rho}_\phi$, we learn the model parameters $\psi$ and $\phi$ by minimizing the following loss functions.

$$\mathcal{L}_{\text{threshold}}(\psi, \phi) = \text{BCE}(\hat{\rho}^{\text{prob}^{(t-1)}}_\Psi, \hat{\rho}^{(t)}_\psi) + \text{BCE}(\hat{\rho}^{\text{prob}^{(t-1)}}_\Phi, \hat{\rho}^{(t)}_\phi), \tag{12}$$

$$\mathcal{L}_{\text{prob}}(\Psi, \Phi) = \text{BCE}(I^{\text{feas}}_{S(n,\hat{\rho}_\psi)}, \hat{\rho}^{\text{prob}}_\Psi) + \text{BCE}(I^{\text{LP-sat}}_{S(n,\hat{\rho}_\pi)}, \hat{\rho}^{\text{prob}}_\Phi) \tag{13}$$

Minimizing $\mathcal{L}_{\text{threshold}}$ and $\mathcal{L}_{\text{prob}}$ imply that $\hat{\rho}_\psi$ and $\hat{\rho}_\phi$ converge to the threshold coverages induced by the indicator random variables $I^{\text{feas}}_{S(n,\hat{\rho}_\psi)}$ and $I^{\text{LP-sat}}_{S(n,\hat{\rho}_\pi)}$, respectively. Our final objective is to minimize the total loss $\mathcal{L}$, which comprises the three components.

$$\mathcal{L} = \sum_t (\mathcal{L}_{\text{coverage}} + \mathcal{L}_{\text{threshold}} + \mathcal{L}_{\text{prob}}) \tag{14}$$

Note that the loss function in equation (14) is fully differentiable.

**Coverage-probability mappings** In equation (12) and (13), $\hat{\rho}^{\text{prob}}_\Psi$ is an intermediary between $\hat{\rho}_\psi$ and $I^{\text{feas}}_{S(n,\hat{\rho}_\Phi)}$. Similarly, $\hat{\rho}^{\text{prob}}_\Phi$ buffers between $\hat{\rho}_\phi$ and $I^{\text{LP-sat}}_{S(n,\hat{\rho}_\pi)}$. In practice, $\hat{\rho}^{\text{prob}}_\Psi$ and $\hat{\rho}^{\text{prob}}_\Phi$ prevent abrupt update of $\hat{\rho}_\psi$ and $\hat{\rho}_\phi$, depending on $I^{\text{feas}}_{S(n,\hat{\rho}_\Phi)} \in \{0,1\}$ and $I^{\text{LP-sat}}_{S(n,\hat{\rho}_\pi)} \in \{0,1\}$, respectively. Let $\rho^{\text{prob}}_\Psi$ be a ground-truth probability that the partial variable assignments $S(n, \rho_\psi)$ is feasible. Let $\rho^{\text{prob}}_\Phi$ be a ground-truth probability that the partial variable assignments $S(n, \rho_\phi)$ satisfies the LP-relaxation solution objective criterion. If $I^{\text{feas}}_{S(n,\hat{\rho}_\Phi)}$ and $I^{\text{LP-sat}}_{S(n,\hat{\rho}_\pi)}$ are non-trivial, there always exist $\rho_\psi \in (0,1)$ and $\rho_\phi \in (0,1)$, such that $\rho_\psi = \rho^{\text{prob}}_\Psi$ and $\rho_\phi = \rho^{\text{prob}}_\Phi$, where $\rho_\psi$ and $\rho_\phi$ are the threshold coverages. Also, if the value of the indicator random variables $I^{\text{feas}}_{S(n,\rho_\psi)}$ and $I^{\text{LP-sat}}_{S(n,\rho_\pi)}$ abruptly shift based on $\rho_\psi$ and $\rho_\pi$, the threshold coverages that correspond to their corresponding probabilities, such that $\rho_\psi = \rho^{\text{prob}}_\Psi$ and $\rho_\phi = \rho^{\text{prob}}_\Phi$, are justified.

**Algorithm** Algorithm 1 describes the overall procedure of TaL. We assume that there is a trained ND model $p_\theta$ to predict the discrete variable values. In Algorithm 1, ThresholdAwareGNN at line 6 extends the model $p_\theta$ to learn $\hat{\rho}_\psi, \hat{\rho}_\phi, \hat{\rho}_\pi$ in equation (8). ThresholdAwareProbGNN at line 7 extends the model $p_\theta$ to learn $\hat{\rho}^{\text{prob}}_\Psi, \hat{\rho}^{\text{prob}}_\Phi$ in equation (9). ThresholdSolve at line 7 computes the feasibility of the variable assignments from $S(n, \hat{\rho}_\psi)$, LP-satisfiability induced from $S(n, \hat{\rho}_\pi)$, and the high-quality feasible solution coverage $\rho^*$, using the MIP solver (See Algorithm 2 in Appendix A.6 for detail). Note that we do not update the ND weight parameters of $p_\theta$ in ThresholdAwareGNN and ThresholdAwareProbGNN.

## 3.2 Theoretical Results on Coverage Optimization in Threshold-aware Learning

In this section, we aim to justify our method by formulating the feasibility property $\mathcal{P}$, LP-relaxation satisfiability property $\mathcal{Q}_\kappa$, and their intersection property $\mathcal{F}_\kappa$. Specifically, we show that there exist two local threshold functions of $\mathcal{F}_\kappa$ that restrict the search space for the optimal coverage $\rho^*$. Finally, we suggest that finding the optimal coverage shows an improved time complexity at the global optimum of $\mathcal{L}_{\text{threshold}}$.

**Threshold functions in MIP variable assignments** We follow the convention in (Bollobás & Thomason, 1987). See Appendix 6 for formal definitions and statements. We leverage Bollobás & Thomason (1987, Theorem 4), which states that every monotone increasing non-trivial property has a threshold function. Given an MIP problem $M = (\mathbf{A}, \mathbf{b}, \mathbf{c})$, we compute $\mathbf{x}$ from the discrete variable value prediction model $p_\theta$. We denote $a \ll b$ or $a(n) \ll b(n)$ when we refer to $\lim_{n\to\infty} a(n)/b(n) = 0$.

**Feasibility of partial variable assignments** We formulate the feasibility property to show that the property has a threshold function. The threshold function in the feasibility property is a criterion that

---

**Algorithm 1** ThresholdAwareLearning

    **Input:** Batch size $H$, the number of iterations $T$, ND model $p_\theta$
1: ThresholdAwareGNN parameterized by $\psi, \phi, \pi$.
2: ThresholdAwareProbGNN parameterized by $\Psi, \Phi$.
3: $i \leftarrow 0$
4: **for** $i \leq T$ **do**
5:     **for** $j \leq H$ **do**
6:         $\mathbf{x}, \hat{\rho}_\psi, \hat{\rho}_\phi, \hat{\rho}_\pi \leftarrow$ ThresholdAwareGNN($M$)
7:         $\hat{\rho}_\Psi^{\text{prob}}, \hat{\rho}_\Phi^{\text{prob}} \leftarrow$ ThresholdAwareProbGNN($M, \hat{\rho}_\psi, \hat{\rho}_\phi, \hat{\rho}_\pi$)
8:         MIPOutput $\leftarrow$ ThresholdSolve($M, \mathbf{x}, \hat{\rho}_\psi, \hat{\rho}_\phi, \hat{\rho}_\pi$)
9:         $I_{S(n,\hat{\rho}_\psi)}^{\text{feas}}, I_{S(n,\hat{\rho}_\pi)}^{\text{LP-sat}}, \rho^* \leftarrow$ MIPOutput
10:        **if** $\rho^*$ is Null **then**
11:           **break**
12:        **end if**
13:        Compute $\mathcal{L}_{\text{coverage}}, \mathcal{L}_{\text{threshold}}, \mathcal{L}_{\text{prob}}$ by eq. equation 11, equation 12, equation 13
14:        $\boldsymbol{g} \leftarrow \nabla_\pi \mathcal{L}_{\text{coverage}}, \nabla_{\psi,\phi} \mathcal{L}_{\text{threshold}}, \nabla_{\Psi,\Phi} \mathcal{L}_{\text{prob}}$
15:        $\boldsymbol{g}_{\text{coverage}}, \boldsymbol{g}_{\text{threshold}}, \boldsymbol{g}_{\text{prob}} \leftarrow \boldsymbol{g}$
16:        Update $\pi, (\psi, \phi), (\Psi, \Phi)$ by gradient descent with $\boldsymbol{g}_{\text{coverage}}, \boldsymbol{g}_{\text{threshold}}, \boldsymbol{g}_{\text{prob}}$
17:        $j \leftarrow j + 1$
18:     **end for**
19:     $i \leftarrow i + 1$
20: **end for**
21: **return** ThresholdAwareGNN

---

depends on both the learned ND model $p_\theta$ and the input problem, used to determine a coverage for generating a feasible solution. We assume $M$ has a feasible solution and the full solution $\mathbf{x}$ generated by the ND model $p_\theta$ is not feasible.

We denote $S(n, \rho) \in \mathcal{P}$ if there exists a feasible solution that contains the corresponding variable values in $S(n, \rho) \subset [n]$. If $\mathcal{P}$ is a nontrivial monotone decreasing property, we can define threshold function $p_0$ of $\mathcal{P}$ as

$$\mathbb{P}(S(n, \rho) \in \mathcal{P}) \to \begin{cases} 1 & \text{if } \rho \ll p_0 \\ 0 & \text{if } \rho \gg p_0 \end{cases} \tag{15}$$

Here, $\mathcal{P}$ is monotone decreasing. Therefore, $\mathcal{P}$ has a threshold function $p_0$ by Theorem A.5 and Lemma A.8 in Appendix A.4.

**LP-relaxation satisfiability of partial variable assignments** We formulate the LP-relaxation satisfiability property to show there exists a local threshold function. Given the learned ND model, we set $\kappa$ to represent the lower bound of the sub-MIP's primal bound. The local threshold function is a criterion depends on the learned ND model and input problem, used to determine a coverage for generating a feasible solution with an objective value of at least $\kappa$.

Let $\hat{\mathbf{x}}(\rho)$ be the LP-relaxation solution values after assigning variable values in $S(n, \rho)$. We denote $S(n, \rho) \in \mathcal{Q}_\kappa$ if $\hat{\mathbf{x}}(\rho)$ is feasible and $\mathbf{c}^\top \hat{\mathbf{x}}(\rho)$ is at least $\kappa$. We define the local threshold function $q_{0,\kappa}$ as

$$\mathbb{P}(S(n, \rho) \in \mathcal{Q}_\kappa) \to \begin{cases} 0 & \text{if } \rho \ll q_{0,\kappa} \ll i \\ 1 & \text{if } i \gg \rho \gg q_{0,\kappa} \end{cases} \tag{16}$$

Here, $i$ is the upper bound of $\rho$. We assume $M$ has a feasible solution, and the full solution $\mathbf{x}$ generated by the ND model $p_\theta$ is not feasible. Also, we assume LP-feasible partial variable assignments exist, such that $\mathcal{Q}_\kappa$ is nontrivial.

If $\mathcal{P}$ is nontrivial and $\kappa$ is fixed, then $\mathcal{Q}_\kappa$ is bounded monotone increasing. By the assumption, $\mathcal{P}$ is nontrivial. $\mathcal{Q}_\kappa$ has a local threshold function by Theorem A.7 and Lemma A.9 in Appendix A.4.

**Feasibility and LP-relaxation satisfiability of partial variable assignments** We formulate the property $\mathcal{F}_\kappa$ as an intersection of the property $\mathcal{P}$ and $\mathcal{Q}_\kappa$. Theorem A.11 in Appendix A.4 states that $\mathcal{F}_\kappa$ has two local thresholds $p_a$, and $p_b$, such that $p_a \ll \rho \ll p_b$ implies that $S(n, \rho)$ satisfies the MIP feasibility and the LP-relaxation objective criterion.

Formally, given that $\mathcal{P}$ is nontrivial monotone decreasing and $\mathcal{Q}_\kappa$ is nontrivial bounded monotone increasing,

$$\mathcal{F}_\kappa := \mathcal{P} \cap \mathcal{Q}_\kappa \tag{17}$$

For some property $\mathcal{H}_n$, a family of subsets $\mathcal{H}_n$ is convex if $B_a \subseteq B_b \subseteq B_c$ and $B_a, B_c \in \mathcal{H}_n$ imply $B_b \in \mathcal{H}_n$. If we bound the domain of $\mathcal{F}_\kappa$ with $p_0$, such that $\rho \in (0, p_0)$ in $S(n, \rho)$ for $\mathcal{F}_\kappa$, then $\mathcal{F}_\kappa$ is convex in the domain, since the intersection of an increasing and a decreasing property is a convex property (Janson et al., 2011). We define $\Delta$-optimal solution $\mathbf{x}_\Delta \in \mathbb{Z}^n$, such that $\mathbf{c}^\top \mathbf{x}_\Delta \leq \mathbf{c}^\top \mathbf{x}^\star + \Delta$. Let $\Delta$-optimal coverage $\rho_\Delta^*$ be the coverage, such that $\mathbf{c}^\top \mathbf{x}(\tau, \rho_\Delta^*) \leq \mathbf{c}^\top \mathbf{x}_\Delta$. If $B(n) \in \mathcal{F}_{\kappa^*}$, we assume that the variable values in $B(n) \subset [n]$ lead to $\Delta$-optimal solution given $\mathbf{x}$. Let $\mathcal{A}$ be the original search space to find the $\Delta$-optimal coverage $\rho_\Delta^*$. Let $t$ be the complexity to find the optimal point as a function of search space size $|\mathcal{A}|$, such that the complexity to find the optimal coverage is $O(t(|\mathcal{A}|))$. Let $\kappa^* := \max \kappa$ subject to $\mathbf{c}^\top \hat{\mathbf{x}}(\rho) = \kappa$ and $\mathbb{P}(S(n, \rho) \in \mathcal{Q}_\kappa) = \xi$ for some $\rho \in (0, 1)$. We assume $\kappa^*$ is given as ground truth in theory, while we adjust $\kappa$ algorithmically in practice. We refer to $p_1(n) \lesssim p_2(n)$ if there exists a positive constant $C$ independent of $n$, such that $p_1(n) \leq C p_2(n)$, for $n > 1/\epsilon$. We assume $q_{0,\kappa} \lesssim p_0$.

We show that TaL reduces the complexity to find the $\Delta$-optimal coverage $\rho_\Delta^*$ by restricting the search space, such that $\rho_\Delta^* \in (q_{0,\kappa^*}, p_0)$, if the test dataset and the training dataset is *i.i.d.* and the time budget to find the feasible solution is sufficient in the test phase.

**Theorem 3.1.** *If the variable values in $S(n, \rho) \in \mathcal{F}_{\kappa^*}$ lead to a $\Delta$-optimal solution, then the complexity of finding the $\Delta$-optimal coverage is $O(t((p_0 - q_{0,\kappa^*})|\mathcal{A}|))$ at the global optimum of $\mathcal{L}_{threshold}$.*

*Proof Sketch.* First, we prove the following statement. If the variable values in $S(n, \rho) \in \mathcal{F}_{\kappa^*}$ lead to $\Delta$-optimal solution and $\rho_\Delta^* = \arg\min_\rho \mathbf{c}^\top \mathbf{x}(\tau, \rho)$, then

$$q_{0,\kappa^*} \lesssim \rho_\Delta^* \lesssim p_0 \tag{18}$$

Finally, we show that $\hat{\rho}_\psi \approx p_0$ and $\hat{\rho}_\phi \approx q_{0,\kappa^*}$ at the global optimum of $\mathcal{L}_{\text{threshold}}$. See the full proof in the Appendix A.4. $\qquad \square$

**Connection to the methodology** At line 8 in Algorithm 1 (ThresholdSolve procedure; details in Appendix A.6), we search for the $\Delta$-optimal coverage using derivative-free optimization (DFO), such as Bayesian optimization or ternary search in the search space restricted into $(\hat{\rho}_\psi, \hat{\rho}_\phi)$. Our GNN outputs converge to the local threshold functions, such that $\hat{\rho}_\psi \approx p_0$ and $\hat{\rho}_\phi \approx q_{0,\kappa^*}$, at the global minimum of the loss function in (12) after iterating over the inner loop of Algorithm 1. It follows that the search space for $\rho_\Delta^*$ becomes $(q_{0,\kappa^*}, p_0)$ at the global minimum of the loss function in (12), by Theorem 3.1. By Theorem 3.1 and our assumption, the cardinality of the search space to find $\rho_\Delta^*$ becomes $(p_0 - q_{0,\kappa^*})|\mathcal{A}|$. Therefore, the post-training complexity of TaL is $O((p_0 - q_{0,\kappa^*})n)$ in Table 1.

**Time complexity** We assume that there are $n$ discrete variables present in each MIP instance within both the training and test datasets. For simplicity, we denote that the cost of training or evaluating a single model is $O(1)$. In the worst-case scenario, ND requires training $n$ models and evaluating all $n$ trained models to find the optimal coverage. Therefore, the training and evaluation of ND require a time complexity of $O(n)$, where training is computationally intensive on GPUs, and evaluation is computationally intensive on CPUs. By excluding the SelectiveNet from ND, CF remarkably improves the training complexity of ND from $O(n)$ to $O(1)$.

In Table 1, CF still requires $O(n)$ time for evaluation in order to determine the optimal cutoff value for the confidence score. TaL utilizes a post-training process as an intermediate step between training and evaluation, thereby reducing the evaluation cost to $O(1)$. For simplicity, we denote $O(1)$ for the MIP solution verification and neural network weight update for each coverage in post-training.

Table 1: Worst-case computational complexity comparison over the methods

|  | Training (GPU ↑) | Post-training (CPU ↑, GPU ↓) | Evaluation (CPU ↑) |
|---|---|---|---|
| ND (Nair et al.) | $O(n)$ | - | $O(n)$ |
| CF (Ours) | $O(1)$ | - | $O(n)$ |
| TaL (Ours) | $O(1)$ | $O((p_0 - q_{0,\kappa^*})n)$ | $O(1)$ |

## 4 Other Related Work

There have been SL approaches to accelerate the primal solution process (Xavier et al., 2021; Nair et al., 2020; Sonnerat et al., 2021; Shen et al., 2021; Ding et al., 2020; Khalil et al., 2022). Empirically, these methods outperform the conventional solver-tuning approaches. Meanwhile, reinforcement learning (RL) is one of the mainstream frameworks for solving CO problems (Khalil et al., 2017; Kool et al., 2018; Kwon et al., 2020; Barrett et al., 2020; Chen & Tian, 2019; Delarue et al., 2020; Lu et al., 2019; Li et al., 2021; Cunha et al., 2018; Manchanda et al., 2019). RL-based approaches represent the actual task performance as a reward in the optimization process. However, the RL reward is non-differentiable and the RL sample complexity hinders dealing with large-scale MIP problems. On the other hand, an unsupervised learning framework to solve CO problems on graphs proposed by Karalias & Loukas (2020) provides theoretical results based on probabilistic methods. However, the framework in Karalias & Loukas (2020) does not leverage or outperform the MIP solver. Among heuristic algorithms, the RENS (Berthold, 2014) heuristic combines two types of methods, relaxation and neighborhood search, to efficiently explore the search space of a mixed integer non-linear programming (MINLP) problem. One potential link between our method and the RENS heuristic is that our method can determine the coverage that leads to a higher success rate, regardless of the (N)LP relaxation used in the RENS algorithm. To the best of our knowledge, our work is the first to leverage the threshold function theory for optimizing the coverage to reduce the discrepancy between the ML and CO problems objectives.

## 5 Experiments

### 5.1 Metrics

We use Primal Integral (PI), as proposed by Berthold (2013) to evaluate the solution quality over the time dimension. PI measures the area between the primal bound and the optimal solution objective value over time. Formally,

$$\text{PI} = \int_{\tau=0}^{T} \mathbf{c}^\top \mathbf{x}^\star(\tau)\mathrm{d}\tau - T\mathbf{c}^\top \mathbf{x}^\star, \tag{19}$$

where $T$ is the time limit, $\mathbf{x}^\star(\tau)$ is the best feasible solution by time $\tau$, and $\mathbf{x}^\star$ is the optimal solution. PI is a metric representing the solution quality in the time dimension, *i.e.,* how fast the quality improves. The optimal instance rate (OR) in Table 2 refers to the average percentage of the test instances solved to optimality in the given time limit. If the problems are not optimally solved, we use the best available solution as a reference to the optimal solution. The optimality gap (OG) refers to the average percentage gap between the optimal solution objective values and the average primal bound values in the given time limit.

### 5.2 Data and evaluation settings

We evaluate the methods with the datasets introduced from (Gasse et al., 2022; 2019). We set one minute to evaluate for relatively hard problems from (Gasse et al., 2022): Workload apportionment and maritime inventory routing problems. For easier problems from (Gasse et al., 2019), we solve for 1, 10, and 5 seconds for the set covering, capacitated facility flow, and maximum independent set problems, respectively. The optimality results in the lower part of Table 2 are obtained by solving the problems using SCIP for 3 minutes, 2

hours, and 10 minutes for the set covering, capacitated facility flow, and maximum independent set problems, respectively. We convert the objective of the capacitated facility flow problem into a minimization problem by negating the objective coefficients, in which the problems are originally formulated as a maximization problem in (Gasse et al., 2019). Furthermore, we measure the performance of the model trained on the maritime inventory routing dataset augmented with MIPLIB 2017 (Gleixner et al., 2021) to verify the effectiveness of data augmentation. Additional information about the dataset can be found in Appendix A.7.

## 5.3 Results

Table 2: Overall results comparing optimality, SCIP, Neural diving, GNNExplainer, Confidence Filter, and Threshold-aware Learning in five MIP datasets: Workload apportionment, maritime inventory routing, set covering, capacitated facility flow, and maximum independent set. Given the time limit, we use Primal Integral (PI) to evaluate the solution quality over time, Primal Bound (PB) to measure the final solution quality, Optimality Gap (OG) to compare the solution quality with the optimal solution, and Optimal instance Rate (OR) to show how many problems are solved to optimality.

| | Workload Apportionment | | | | Maritime Inventory Routing (non-augmented) | | | | Maritime Inventory Routing (augmented) | | | |
|---|---|---|---|---|---|---|---|---|---|---|---|---|
| | PI ↓ | PB ↓ | OG (%) ↓ | OR (%) ↑ | PI ↓ | PB ↓ | OG (%) ↓ | OR (%) ↑ | PI ↓ | PB ↓ | OG (%) ↓ | OR (%) ↑ |
| SCIP (30 hrs) | - | 708.31 | 0.01 | 98 | - | 50175.95 | 0 | 100 | - | 50175.95 | 0 | 100 |
| SCIP (1 min) | 48182.31 | 738.85 | 4.36 | 0 | 41682758.55 | 647961.18 | 305.48 | 30 | 41682758.55 | 647961.18 | 305.48 | 30 |
| Neural diving (Nair et al., 2020) | 44926.06 | 713.10 | 0.69 | 2 | 28357794.50 | 372770.44 | 143.08 | 30 | 30101028.39 | 322748.95 | 115.88 | 30 |
| GNNExplainer (Ying et al., 2019) | 47165.63 | 719.96 | 1.65 | 0 | 29343874.24 | 369122.56 | 141.50 | 20 | 39004703.50 | 509108.16 | 249.95 | 25 |
| Confidence Filter (Ours) | 44862.77 | 711.43 | 0.47 | 2 | 24581204.50 | 202526.50 | 83.91 | **35** | **24795796.07** | 306165.57 | 136.55 | 30 |
| Threshold-aware Learning (Ours) | **44634.85** | **711.34** | **0.45** | **4** | **24288490.56** | **192074.00** | **69.97** | 30 | 33639950.16 | **287189.69** | 106.68 | **35** |

| | Set Covering | | | | Capacitated Facility Flow | | | | Maximum Independent Set | | | |
|---|---|---|---|---|---|---|---|---|---|---|---|---|
| | PI ↓ | PB ↓ | OG (%) ↓ | OR (%) ↑ | PI ↓ | PB ↓ | OG (%) ↓ | OR (%) ↑ | PI ↓ | PB ↓ | OG (%) ↓ | OR (%) ↑ |
| Optimality | − | 225.79 | 0 | 100 | − | 18049.77 | 0 | 100 | − | −226.61 | 0 | 100 |
| SCIP | 788.04 | 606.68 | 165.96 | 2 | 834198.50 | 43172.72 | 139.40 | 0 | −1009.56 | −218.64 | 3.55 | 10 |
| Neural diving (Nair et al., 2020) | 493.65 | 246.12 | 8.99 | 0 | **312589.87** | 18221.34 | 0.93 | **34** | −1062.37 | −226.51 | 0.05 | 95 |
| Confidence Filter (Ours) | 431.31 | 233.06 | 3.13 | **5** | 319075.00 | 18214.33 | 0.91 | 16 | **−1081.87** | −226.44 | 0.08 | 87 |
| Threshold-aware Learning (Ours) | **411.45** | **232.99** | **3.09** | 3 | 318984.36 | **18147.82** | **0.54** | 23 | −1075.95 | **−226.57** | **0.02** | **96** |

Table 2 shows that TaL outperforms the other methods on all five datasets. Also, CF outperforms ND in relatively hard problems: Workload apportionment and maritime inventory routing problems. In the workload apportionment dataset, TaL shows a 0.45% optimality gap at the one-minute time limit, roughly 10× better than SCIP. In the maritime inventory routing non-augmented dataset, TaL achieves a 70% optimality gap, which is 2× better than ND and 3× better than SCIP. In relatively easy problems, TaL outperforms ND in the optimality gap by 3×, 2×, and 2× in the set covering, capacitated facility flow, and maximum independent set problems, respectively. On average, TaL takes 10 seconds to show 0.54% optimality gap while SCIP takes 2 hours to solve capacitated facility flow test problems to optimality. The best PI value (TaL) in the data-augmented maritime inventory routing dataset is close to the best PI value (CF) in the non-augmented dataset. Moreover, CF in the data-augmented dataset solves the same number of instances to the optimality as the TaL in the non-augmented dataset.

## 6 Conclusion

In this work, we present the CF and TaL methods to enhance the learning-based MIP optimization performance. We propose a provably efficient learning algorithm to estimate the search space for coverage optimization. We also provide theoretical justifications for learning to optimize coverage in generating feasible solutions for MIP, from the perspective of probabilistic combinatorics. Finally, we empirically demonstrate that the variable assignment coverage is an effective auxiliary measure to fill the gaps between the SL and MIP objectives, showing competitive results against other methods.

One limitation of our approach is that the neural network model relies on collecting MIP solutions, which can be computationally expensive. Hence, it would be intriguing to devise a novel self-supervised learning

approach to train a foundational model for solving MIP problems in combination with TaL. Also, another direction for future work is to extend TaL to a broader area of machine learning in a high-dimensional setting.

**Author Contributions**

**Acknowledgments**

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

# A Appendix

## A.1 Notations

- $n$: the number of variables

- $m$: the number of linear constraints

- $r$: the number of discrete variables

- $\mathbf{x} = [x_1, \ldots, x_n] \in \mathbb{R}^n$: the vector of variable values

- $\mathbf{x}_{\text{int}} \in \mathbb{Z}^r, \mathbf{x}_{\text{cont}} \in \mathbb{R}^{n-r}$ and $\mathbf{x} = [\mathbf{x}_{\text{int}}; \mathbf{x}_{\text{cont}}]$

- $\mathbf{x}^\star$: the vector of optimal solution variable values

- $\mathbf{x}(\tau, \rho)$: the vector of primal solution at time $\tau$ with initial variable assignments of $\mathbf{x}$ with coverage $\rho$

- $\hat{\mathbf{x}}(\rho)$: the LP-relaxation solution with initial variable assignments of $\mathbf{x}$ with coverage $\rho$

- $\bar{\mathbf{x}}$: the vector of LP-relaxation solution variable values

- $\mathbf{c} \in \mathbb{R}^n$: the vector of objective coefficients

- $\mathbf{A} \in \mathbb{R}^{m \times n}$: the linear constraint coefficient matrix

- $\mathbf{b} \in \mathbb{R}^m$: the vector of linear constraint upper bound

- $\mathbf{e}_i = \mathbf{e}_i^{(n)} \in \mathbb{R}^n, i = 1, \ldots, n$: the Euclidean standard basis

- $[n] = \{1, 2, \ldots, n\}$: the index set

- $M = (\mathbf{A}, \mathbf{b}, \mathbf{c})$: the MIP problem

- $\mathcal{I}$: the set of dimensions of $\mathbf{x}$ that is discrete

- $x_d$: the $d$-th dimension of $\mathbf{x}$

- $s_d$: the binary classifier output to decide to fix $x_d$

- $p_\theta$: Neural diving model (without SelectiveNet) parameterized by $\theta$

- $\Gamma$: the cutoff value for Post hoc Confidence Filter

- $\mathcal{R}(M)$: the set of feasible solution(s) $\mathbf{x} \in \mathbb{Z}^r \times \mathbb{R}^{n-r}$

- $\bar{\mathcal{R}}(M)$: the set of LP-relaxation solution(s) $\mathbf{x} \in \mathbb{R}^n$ which satisfies $\mathbf{A}\mathbf{x} \le \mathbf{b}$ regardless the integrality constraints

- $\hat{\mathcal{R}}(M)$: the set of LP-feasible solution(s) $\mathbf{x} \in \mathbb{Z}^r \times \mathbb{R}^{n-r}$ which satisfies $\mathbf{A}\mathbf{x} \le \mathbf{b}$ after partially fixing discrete variables of $\mathbf{x}$

- $\xi \in (0, 1)$: the probability at which threshold behavior occurs

- $\mathscr{P}(X)$: the power set of $X$

- $a \ll b$ or $a(n) \ll b(n) : \lim_{n \to \infty} a(n)/b(n) = 0$

- $a \gg b$ or $a(n) \gg b(n) : \lim_{n \to \infty} a(n)/b(n) = \infty$

- $a \lesssim b : a(n) \lesssim b(n) :$ there exists $C > 0$, s. t. $a(n) \le Cb(n)$, where $n \in \mathcal{N}$, for some set $\mathcal{N}$

- $\kappa$: the target objective value of the LP-solution for partial discrete variables fixed

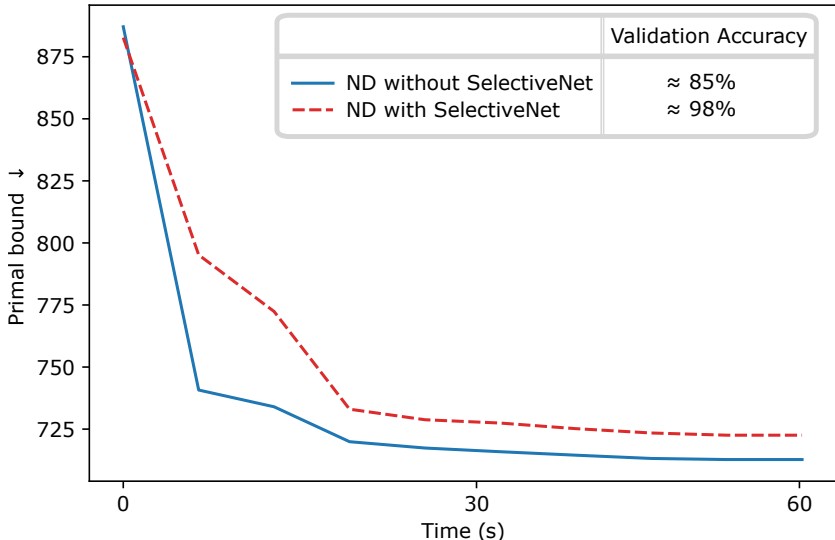

Figure 3: The variable value classification accuracy and the primal bound comparison in the validation dataset according to the existence of SelectiveNet. Variable value classification accuracy measures how many integer variables values match out of the total integer variables between the neural network output and the solution; the higher, the better. The primal bound represents the MIP solution quality; the lower, the better. We measure the primal bound over time at the one-minute time limit. The average discrete variable assignment rate is 20% for both models. We use Confidence Filter for 'ND without SelectiveNet'. SelectiveNet improves the classification accuracy by roughly 13 percentage points margin, whereas Neural diving without SelectiveNet outperforms SelectiveNet in the primal bound. The target dataset is the Workload Apportionment problem from Gasse et al. (2022).

### A.2 Discrepancy between the learning and MIP objectives

The ND learning objective is to minimize the loss function formulated as a negative log likelihood. On the other hand, MIP aims to optimize the objective function satisfying the constraints. Let's say the learning objective is denoted by minimize $\mathcal{L}$. Let's say the MIP objective is denoted by minimize $\mathbf{c}^\top \mathbf{x}$. Assume that we obtain the solution $\tilde{\mathbf{x}}$, which can be infeasible, from the ND model $p_\theta(\mathbf{x}|M)$ that corresponds to $\mathcal{L}$. Let $\tilde{\mathbf{x}}(\tau, \rho)$ be a feasible solution at solving time $\tau$ with initial variable assignment of $\tilde{\mathbf{x}}$ with coverage $rho$. It follows that $\mathbf{c}^\top \tilde{\mathbf{x}}(\tau, \rho)$ is not monotone with respect to its corresponding loss $\mathcal{L}$. In Figure 3, ND with SelectiveNet outperforms ND without SelectiveNet by roughly 13 percentage points in validation variable classification accuracy. In contrast, ND without SelectiveNet outperforms ND with SelectiveNet in the primal bound.

### A.3 Formal Definitions

Let $\mathscr{P}([n])$ denotes the power set of $[n] = \{1, \ldots, n\}$. A *nontrivial property* $\mathcal{H}_n$ is a nonempty collection of subsets of $[n]$, where the subsets satisfy the conditions of $\mathcal{H}_n$ and $\mathcal{H}_n \neq \mathscr{P}([n])$ . We call $\mathcal{H}_n$ *monotone increasing* if $B_{ss} \in \mathcal{H}_n$ and $B_{ss} \subset B_s \subset [n]$ implies $B_s \in \mathcal{H}_n$. We call $\mathcal{H}_n$ *monotone decreasing* if $B_s \in \mathcal{H}_n$ and $B_{ss} \subset B_s$ implies $B_{ss} \in \mathcal{H}_n$. Note that a monotone increasing (decreasing) property $\mathcal{H}_n$ is non-trivial *if and only if* $\emptyset \notin \mathcal{H}_n$ and $[n] \in \mathcal{H}_n$ ($\emptyset \in \mathcal{H}_n$ and $[n] \notin \mathcal{H}_n$).

For $i = 0, \ldots, n$, let $\mathcal{B}_i \subset [n]$ denote the collection of subsets of $[n]$ with $i$ elements, *i.e.* $|\mathcal{B}_i| = \binom{n}{i}$, and define $\mathcal{H}_i := \mathcal{H}_n \cap \mathcal{B}_i$. Note that

$$\mathbb{P}(\mathcal{H}_n|\mathcal{B}_i) = \frac{|\mathcal{H}_i|}{|\mathcal{B}_i|} = \frac{|\mathcal{H}_i|}{\binom{n}{i}} \tag{20}$$

where the conditional probability $\mathbb{P}(\mathcal{H}_n|\mathcal{B}_i)$ denotes the probability that a random set of $\mathcal{B}_i$ has property $\mathcal{H}_n$. If an increasing sequence $m = m(n)$ satisfies that $\mathbb{P}(\mathcal{H}_n|\mathcal{B}_{m(n)}) \to 1$ as $n \to \infty$, then we say that $m$-subset of $\mathcal{B}_{m(n)}$ has $\mathcal{H}_n$ almost surely. Similarly, we say that $m$-subset of $\mathcal{B}_{m(n)}$ fails to have $\mathcal{H}_n$ almost surely if $\mathbb{P}(\mathcal{H}_n|\mathcal{B}_{m(n)}) \to 0$ as $n \to \infty$.

**Definition A.1.** A function $m^*(n)$ is a threshold function for a monotone increasing property $\mathcal{H}_n$ if for $m/m^* \to 0$ as $n \to \infty$ (we write $m \ll m^*$), $m$-subset of $\mathcal{B}_n$ fails to have $\mathcal{H}_n$ almost surely and for $m/m^* \to \infty$ as $n \to \infty$ (we write $m \gg m^*$), $m$-subset of $\mathcal{B}_n$ has $\mathcal{H}_n$ almost surely:

$$\lim_{n\to\infty} \mathbb{P}(\mathcal{H}_n|\mathcal{B}_{m(n)}) = \begin{cases} 0 & : m(n) \ll m^*(n) \\ 1 & : m(n) \gg m^*(n). \end{cases} \tag{21}$$

Similarly, a function $m^*(n)$ is said to be a threshold function for a monotone decreasing property $\mathcal{H}_n$ if for $m/m^* \to \infty$ as $n \to \infty$ , $m$-subset of $\mathcal{B}_n$ fails to have $\mathcal{H}_n$ almost surely and for $m/m^* \to 0$ as $n \to \infty$, $m$-subset of $\mathcal{B}_n$ has $\mathcal{H}_n$ almost surely:

$$\lim_{n\to\infty} \mathbb{P}(\mathcal{H}_n|\mathcal{B}_{m(n)}) = \begin{cases} 0 & : m(n) \gg m^*(n) \\ 1 & : m(n) \ll m^*(n). \end{cases} \tag{22}$$

We say $\mathcal{H}_n$ is *bounded monotone decreasing* if $B_s \in \mathcal{H}_n$ and $B_{ss} \subset B_s \in \mathcal{B}_i$ implies $B_{ss} \in \mathcal{H}_n$ for $i < n$. We say $\mathcal{H}_n$ is *bounded monotone increasing* if $B_{ss} \in \mathcal{H}_i$ and $B_{ss} \subset B_s \in \mathcal{B}_i$ implies $B_s \in \mathcal{H}_i$ for $i < n$. We introduce a local version of a threshold function in the Definition A.1 for a bounded monotone increasing and decreasing property.

**Definition A.2.** A function $l^*(n)$ is a local threshold function for a bounded monotone increasing property $\mathcal{H}_n$ if for $l/l^* \to 0, l^*/i \to 0$ as $n \to \infty$, $l$-subset of $\mathcal{B}_n$ fails to have $\mathcal{H}_n$ almost surely, and for $l/l^* \to \infty, i/l \to \infty$ as $n \to \infty$, $l$-subset of $\mathcal{B}_n$ has $\mathcal{H}_n$ almost surely:

$$\lim_{n\to\infty} \mathbb{P}(\mathcal{H}_n|\mathcal{B}_{l(n)}) = \begin{cases} 0 & : l(n) \ll l^*(n) \ll i(n) \\ 1 & : i(n) \gg l(n) \gg l^*(n). \end{cases} \tag{23}$$

Similarly, a function $l^*(n)$ is said to be a local threshold function for a bounded monotone decreasing property $\mathcal{H}_n$ if for $l/l^* \to \infty, i/l \to \infty$ as $n \to \infty$, $l$-subset of $\mathcal{B}_n$ fails to have $\mathcal{H}_n$ almost surely and for $l/l^* \to 0, l^*/i \to 0$ as $n \to \infty$, $l$-subset of $\mathcal{B}_n$ has $\mathcal{H}_n$ almost surely:

$$\lim_{n\to\infty} \mathbb{P}(\mathcal{H}_n|\mathcal{B}_{l(n)}) = \begin{cases} 0 & : i(n) \gg l(n) \gg l^*(n) \\ 1 & : l(n) \ll l^*(n) \ll i(n). \end{cases} \tag{24}$$

From Definition A.2, we have the statement about local threshold functions in bounded monotone properties in Theorem A.7 in Appendix A.4. Theorem A.7 states that every nontrivial bounded monotone increasing property has a local threshold function in a bounded domain.

**Definition A.3.** Given an MIP problem $M = (\mathbf{A}, \mathbf{b}, \mathbf{c})$, $\mathbf{x} \in \mathbb{R}^n$, we define the property $\mathcal{P}$ as

$$\mathcal{P}(\mathbf{x}, \mathbf{A}, \mathbf{b}) := \{B(n) \subset [n] : \text{ there exists } \mathbf{y}' \in \mathbb{Z}^r \times \mathbb{R}^{n-r} \text{ subject to } \mathbf{A}[\mathbf{y} + \mathbf{y}'] \leq \mathbf{b}\}, \tag{25}$$

where $\mathbf{y} = \sum_{i \in B(n)} x_i \mathbf{e}_i$, and $\mathbf{y}' = \sum_{i \in [n]\setminus B(n)} y_i \mathbf{e}_i$.

**Definition A.4.** Given an MIP problem $M = (\mathbf{A}, \mathbf{b}, \mathbf{c})$, $\mathbf{x} \in \mathbb{R}^n$, and $\kappa \in \mathbb{R}$, we define the property $\mathcal{Q}_\kappa$ as

$$\begin{aligned} \mathcal{Q}_\kappa(\mathbf{x}, \mathbf{A}, \mathbf{b}, \mathbf{c}) := \{B(n) \subset [n] : & \text{ there exists } \mathbf{z}' \in \mathbb{R}^n \\ & \text{subject to } \mathbf{c}^\top[\mathbf{z} + \mathbf{z}'] \geq \kappa \text{ and } \mathbf{A}[\mathbf{z} + \mathbf{z}'] \leq \mathbf{b}\}, \end{aligned} \tag{26}$$

where $\mathbf{z} = \sum_{i \in B(n)} x_i \mathbf{e}_i$ and $\mathbf{z}' = \sum_{i \in [n]\setminus B(n)} z_i \mathbf{e}_i$.

### A.4 Theoretical Statements

**Theorem A.5.** *Let $\xi \in (0,1)$. For each $n \in \mathbb{N}$, let $\mathcal{H}_n$ be a monotone increasing non-trivial property and set*

$$m_*(n) := \max \left\{ m : \mathbb{P}(\mathcal{H}_n | \mathcal{B}_m) \leq \xi \right\}. \tag{27}$$

*If $m \leq m_*(n)$, then*

$$\mathbb{P}(\mathcal{H}_n | \mathcal{B}_m) \leq 1 - \xi^{\frac{m}{m_*(n)}} \tag{28}$$

*and, if $m \geq m_*(n) + 1$, then*

$$\mathbb{P}(\mathcal{H}_n | \mathcal{B}_m) \geq 1 - \xi^{\frac{m}{m_*(n)+1}}. \tag{29}$$

*In particular, $m_*(n)$ is a threshold function of $\mathcal{H}_n$.*

*Proof.* We slightly modify the proof in Bollobás & Thomason (1987, Theorem 4) for rigorousness. Let $\mathcal{J}_n$ denote the negation of $\mathcal{H}_n$. Note that $\mathcal{J}_n$ is a nontrivial monotone decreasing property.

If $m \leq m_*(n)$, then

$$\mathbb{P}(\mathcal{J}_n | \mathcal{B}_m) \geq \mathbb{P}(\mathcal{J}_n | \mathcal{B}_{m_*(n)}) \geq \xi^{\frac{m}{m_*(n)}}$$

If $m \geq m_*(n) + 1$, then

$$\mathbb{P}(\mathcal{J}_n | \mathcal{B}_m) \leq \mathbb{P}(\mathcal{J}_n | \mathcal{B}_{m_*(n)+1}) \leq \xi^{\frac{m}{m_*(n)+1}}$$

$\square$

**Corollary A.6.** *Let $\xi \in (0,1)$. For each $n \in \mathbb{N}$, let $\mathcal{H}_n$ be a monotone decreasing nontrivial property.*

$$m_*(n) := \min \left\{ m : \mathbb{P}(\mathcal{H}_n | \mathcal{B}_m) \geq \xi \right\}. \tag{30}$$

*If $m \leq m_*(n)$, then*

$$\mathbb{P}(\mathcal{H}_n | \mathcal{B}_m) \geq 1 - \xi^{\frac{m}{m_*(n)}} \tag{31}$$

*If $m \geq m_*(n) + 1$, then*

$$\mathbb{P}(\mathcal{H}_n | \mathcal{B}_m) \leq 1 - \xi^{\frac{m}{m_*(n)+1}} \tag{32}$$

*In particular, $m_*(n)$ is a threshold function of $\mathcal{H}_n$.*

*Proof.* Let $\mathcal{J}_n$ denote the negation of $\mathcal{H}_n$. Note that $\mathcal{J}_n$ is a nontrivial monotone increasing property.

If $m \leq m_*(n)$, then

$$\mathbb{P}(\mathcal{J}_n | \mathcal{B}_m) \leq \mathbb{P}(\mathcal{J}_n | \mathcal{B}_{m_*(n)}) \leq \xi^{\frac{m}{m_*(n)}}$$

If $m \geq m_*(n) + 1$, then

$$\mathbb{P}(\mathcal{J}_n | \mathcal{B}_m) \geq \mathbb{P}(\mathcal{J}_n | \mathcal{B}_{m_*(n)+1}) \geq \xi^{\frac{m}{m_*(n)+1}}$$

$\square$

**Theorem A.7.** *Let $\xi \in (0,1)$. For each $n \in \mathbb{N}$, let $\mathcal{H}_n$ be a non-trivial bounded monotone increasing property, where the bound is $i$.*

$$l_*(n) := \max \left\{ l : \mathbb{P}(\mathcal{H}_n | \mathcal{B}_l) \leq \xi \right\}. \tag{33}$$

*If $l \geq i + 1$, then it is of limited interest since $\mathcal{H}_n$ is non-monotone in such domain. If $l \leq l_*(n) \leq i$, then*

$$\mathbb{P}(\mathcal{H}_n | \mathcal{B}_l) \leq 1 - \xi^{\frac{l}{l_*(n)}} \quad and \quad \mathbb{P}(\mathcal{H}_n | \mathcal{B}_{l_*(n)}) \leq 1 - \xi^{\frac{l_*(n)}{i}} \tag{34}$$

*If $i \geq l \geq l_*(n) + 1$, then*

$$\mathbb{P}(\mathcal{H}_n | \mathcal{B}_l) \geq 1 - \xi^{\frac{l}{l_*(n)+1}} \quad and \quad \mathbb{P}(\mathcal{H}_n | \mathcal{B}_i) \geq 1 - \xi^{\frac{i}{l}} \tag{35}$$

*In particular, $l_*$ is a local threshold function of $\mathcal{H}_n$ bounded by $i$.*

*Proof.* We adapt the proof in Bollobás & Thomason (1987, Theorem 4) for Definition A.2. Let $\mathcal{J}_n$ denote the negation of $\mathcal{H}_n$. Note that $\mathcal{J}_n$ is a nontrivial bounded monotone decreasing property.

If $l \leq l_*(n) \leq i$, then

$$\mathbb{P}(\mathcal{J}_n | \mathcal{B}_l) \geq \mathbb{P}(\mathcal{J}_n | \mathcal{B}_{l_*(n)}) \geq \xi^{\frac{l}{l_*(n)}} \quad \text{and} \quad \mathbb{P}(\mathcal{J}_n | \mathcal{B}_{l_*(n)}) \geq \mathbb{P}(\mathcal{J}_n | \mathcal{B}_i) \geq \xi^{\frac{l_*(n)}{i}}$$

If $i \geq l \geq l_*(n) + 1$, then

$$\mathbb{P}(\mathcal{J}_n | \mathcal{B}_l) \leq \mathbb{P}(\mathcal{J}_n | \mathcal{B}_{l_*(n)+1}) \leq \xi^{\frac{l}{l_*(n)+1}} \quad \text{and} \quad \mathbb{P}(\mathcal{J}_n | \mathcal{B}_i) \leq \mathbb{P}(\mathcal{J}_n | \mathcal{B}_l) \leq \xi^{\frac{i}{l}}$$

$\square$

**Lemma A.8.** *$\mathcal{P}$ is monotone decreasing.*

*Proof.* Let $B_s$ be a set, such that $B_s \subset B$. Let $B_{ss}$ be a set, such that $B_{ss} \subset B_s$. By definition, if $B_s \in \mathcal{P}$, then there exists $\mathbf{z}'_s$, such that $\mathbf{A}[\mathbf{z}_s + \mathbf{z}'_s] \leq \mathbf{b}$, where $\mathbf{z}_s = \sum_{i \in B_s} x_i \mathbf{e}_i$, and $\mathbf{z}'_s = \sum_{i \in [n] \setminus B_s} z_i \mathbf{e}_i$, given $\mathbf{x}$. Let $\mathbf{z}_{ss} = \sum_{i \in B_{ss}} x_i \mathbf{e}_i$, and $\mathbf{z}'_{ss} = \sum_{i \in [n] \setminus B_{ss}} z_i \mathbf{e}_i$ for $B_{ss} \subset B_s$. We can choose $\mathbf{z}'_{ss}$, such that $\mathbf{z}'_{ss} = \mathbf{z}'_s + \sum_{i \in B_s \setminus B_{ss}} x_i \mathbf{e}_i$, where $\sum_{i \in B_s \setminus B_{ss}} x_i \mathbf{e}_i = \mathbf{z}_s - \sum_{i \in B_{ss}} x_i \mathbf{e}_i$. It follows that there exists $\mathbf{z}'_{ss}$, such that $\mathbf{A}[\mathbf{z}_{ss} + \mathbf{z}'_{ss}] = \mathbf{A}[\mathbf{z}_s + \mathbf{z}'_s] \leq \mathbf{b}$. Hence, $B_s \in \mathcal{P}$ implies $B_{ss} \in \mathcal{P}$. $\square$

**Lemma A.9.** *Fix $\kappa$. If $\mathcal{P}$ is nontrivial, then $\mathcal{Q}_\kappa$ is bounded monotone increasing.*

*Proof.* $\mathcal{P}$ has a threshold function $p_0$ by Theorem A.5 and Lemma A.8. We choose $p_0$ for the bound of $\mathcal{Q}_\kappa$. Therefore, we inspect only the domain of interest $q \in [0, p_0]$ in $S(n, q)$ for $\mathcal{Q}_\kappa$. Let $\mathbf{u}_p = \sum_{i \in B_p} x_i \mathbf{e}_i \in \mathbb{Z}^n$ for $B_p = S(n, p_0) \in \mathcal{P}$. Given $B_p \in \mathcal{P}$, there exists $\mathbf{u}'_p = \sum_{i \in [n] \setminus B_p} u_i \mathbf{e}_i \in \mathbb{R}^n$, such that $\mathbf{A}[\mathbf{u}_p + \mathbf{u}'_p] \leq \mathbf{b}$, since $\mathbb{Z}^n \subset \mathbb{R}^n$.

Let $B_s$ be a set, such that $B_s \subset B_p$. Let $B_{ss}$ be a set, such that $B_{ss} \subset B_s$. By definition, $B_{ss} \in \mathcal{Q}_\kappa$ implies there exists $\mathbf{u}'_{ss} = \sum_{i \in [n] \setminus B_{ss}} u_i \mathbf{e}_i \in \mathbb{R}^n$, such that $\mathbf{A}[\mathbf{u}_{ss} + \mathbf{u}'_{ss}] \leq \mathbf{b}$, and $\mathbf{c}^\top[\mathbf{u}_{ss} + \mathbf{u}'_{ss}] \geq \kappa$ where $\mathbf{u}_{ss} = \sum_{i \in B_{ss}} x_i \mathbf{e}_i \in \mathbb{Z}^n$, given $\mathbf{x}$. Let $\mathbf{u}_s = \sum_{i \in B_s} x_i \mathbf{e}_i \in \mathbb{Z}^n$, and $\mathbf{u}'_s = \sum_{i \in [n] \setminus B_s} u_i \mathbf{e}_i \in \mathbb{R}^n$. We show that 1) $B_{ss} \in \mathcal{Q}_\kappa$ implies $\mathbf{A}[\mathbf{u}_s + \mathbf{u}'_s] \leq \mathbf{b}$, and 2) $B_{ss} \in \mathcal{Q}_\kappa$ implies $\mathbf{c}^\top[\mathbf{u}_s + \mathbf{u}'_s] \geq \kappa$.

1) Let $\mathbf{u}_{s-ss} = \sum_{i \in B_s \setminus B_{ss}} x_i \mathbf{e}_i \in \mathbb{Z}^n$. Let $\mathbf{u}'_p = \sum_{i \in [n] \setminus B_p} u_i \mathbf{e}_i \in \mathbb{R}^n$. For any $B_s \supset B_{ss}$, $B_s \setminus B_{ss} \in \mathcal{P}$. Also, we can find $\mathbf{u}'_s = \mathbf{u}_{p-s} + \mathbf{u}'_p$, such that

$$\mathbf{A}[\mathbf{u}_{ss} + \mathbf{u}_{s-ss} + \mathbf{u}_{p-s} + \mathbf{u}'_p] = \mathbf{A}[\mathbf{u}_s + \mathbf{u}'_s] = \mathbf{A}[\mathbf{u}_p + \mathbf{u}'_p] \leq \mathbf{b}, \tag{36}$$

since $\mathbb{Z}^n \subset \mathbb{R}^n$. Therefore, $\mathbf{A}[\mathbf{u}_s + \mathbf{u}'_s] \leq \mathbf{b}$.

2) Let $\mathcal{R}(M)$ be the set of feasible solutions. Let $\hat{\mathcal{R}}(M)$ be the set of LP-feasible solution $\hat{\mathbf{u}}$, after fixing the nontrivial number of discrete variables of $M$. Let $\bar{\mathcal{R}}(M)$ be the set of LP-feasible solution $\bar{\mathbf{u}}$, without fixing variables of $M$. Since $\mathcal{R}(M) \subseteq \hat{\mathcal{R}}(M) \subseteq \bar{\mathcal{R}}(M)$,

$$\min_{\bar{\mathbf{u}} \in \bar{\mathcal{R}}(M)} \mathbf{c}^\top \bar{\mathbf{u}} \leq \min_{\hat{\mathbf{u}} \in \hat{\mathcal{R}}(M)} \mathbf{c}^\top \hat{\mathbf{u}} \leq \min_{\mathbf{u} \in \mathcal{R}(M)} \mathbf{c}^\top \mathbf{u} \tag{37}$$

Therefore, for any $B_s$,

$$\mathbf{c}^\top [\mathbf{u}_s + \mathbf{u}_s'] = \mathbf{c}^\top [\mathbf{u}_{ss} + \mathbf{u}_{s-ss} + \mathbf{u}_s'] \geq \mathbf{c}^\top [\mathbf{u}_{ss} + \mathbf{u}_{ss}'] \geq \kappa \tag{38}$$

By 1) and 2), $B_{ss} \in \mathcal{Q}_\kappa$ implies $B_s \in \mathcal{Q}_\kappa$. Hence, $\mathcal{Q}_\kappa$ is bounded monotone increasing, in which the bound is at most $p_0$, such that $q \in [0, p_0]$ in $S(n, q)$ for $\mathcal{Q}_\kappa$. $\qquad\square$

**Lemma A.10.** *If $\mathcal{P}$ is nontrivial, then $q_{0,\kappa}$ is monotonic increasing with regard to $\kappa$, bounded by $p_0$.*

*Proof.* Let $\mathbf{x}^\star$ be the vector of optimal solution variable values. Let $\bar{\mathbf{x}}$ be the vector of LP-relaxation solution variable values. By Lemma A.9 and Theorem A.7, $\mathcal{Q}_\kappa$ is bounded monotone increasing and has a local threshold function $q_{0,\kappa}$, where $\kappa$ is fixed. We choose the constant $\xi \in (0, 1)$ as in Theorem A.5, such that $\mathbb{P}(S(n, q_{0,\kappa}) \in \mathcal{Q}_\kappa) = \xi$. We choose $p_0$ as a bound of $\mathcal{Q}_\kappa$ as in Lemma *A.9*. Suppose $\kappa_1 \leq \kappa_2$, such that $\mathbf{c}^\top \bar{\mathbf{x}} < \kappa_1 \leq \kappa \leq \kappa_2 < \mathbf{c}^\top \mathbf{x}^\star$. If follows that

$$\mathbb{P}(S(n, q_{0,\kappa}) \in \mathcal{Q}_{\kappa_1}) \geq \mathbb{P}(S(n, q_{0,\kappa}) \in \mathcal{Q}_\kappa) \geq \mathbb{P}(S(n, q_{0,\kappa}) \in \mathcal{Q}_{\kappa_2}) \tag{39}$$

$\qquad\square$

**Theorem A.11.** *Fix $\kappa$. If $q_{0,\kappa} \ll p_0$, then $\mathcal{F}_\kappa$ has an interval of certainty, such that $q_{0,\kappa} \ll \rho \ll p_0$ implies $\mathbb{P}(S(n, \rho) \in \mathcal{F}_\kappa) \to 1$.*

*Proof.* By Lemma A.8 and A.9, there exist $p_0$ and $q_0$ such that

$$\mathbb{P}(S(n, \rho) \in \mathcal{P}) \to \begin{cases} 1 & \text{if } \rho \ll p_0 \\ 0 & \text{if } \rho \gg p_0 \end{cases}$$

and

$$\mathbb{P}(S(n, \rho) \in \mathcal{Q}_\kappa) \to \begin{cases} 0 & \text{if } \rho \ll q_{0,\kappa} \ll p_0 \\ 1 & \text{if } p_0 \gg \rho \gg q_{0,\kappa} \end{cases}$$

Since

$$\mathbb{P}(S(n, \rho) \in \mathcal{F}_\kappa) \geq \mathbb{P}(S(n, \rho) \in \mathcal{P}) + \mathbb{P}(S(n, \rho) \in \mathcal{Q}_\kappa) - 1 \tag{40}$$

By equation (40),

$$q_{0,\kappa} \ll \rho \ll p_0 \text{ implies } \mathbb{P}(S(n, \rho) \in \mathcal{F}_\kappa) \to 1 \tag{41}$$

$\qquad\square$

**Proposition A.12.** *Fix $\hat{\rho}_\phi$ and $\hat{\rho}_\Phi^{prob}$. If $\hat{\rho}_\psi \sim Uniform(0, 1)$. then the optimal $\hat{\rho}_\psi$ in $\mathcal{L}_{threshold}$ is $p_0$.*

*Proof.* By Theorem A.5 and Lemma A.8, $\mathcal{P}$ has a threshold function $p_0$. Suppose there exist $H$ samples. For $\hat{\rho}_\phi$ and $\hat{\rho}_\Phi^{\text{prob}}$ fixed, the training of $\mathcal{L}_{\text{prob}}$ is to maximize

$$\sum_{i=1}^{H} I_{S(n,\rho) \in \mathcal{P}}^{(i)} \log \hat{\rho}_\Psi^{\text{prob}} + \sum_{i=1}^{H} (1 - I_{S(n,\rho) \in \mathcal{P}}^{(i)}) \log(1 - \hat{\rho}_\Phi^{\text{prob}}) \tag{42}$$

For any $(a, b) \in \mathbb{R}^2 \setminus (0, 0)$, let $Z : w \to a \log(w) + b \log(1 - w)$. Then $\arg\max Z(w) = \frac{a}{a+b}$. Hence,

$$\arg\min \mathcal{L}_{\text{prob}} = \frac{1}{H} \sum_{i=1}^{H} I_{S(n,\rho) \in \mathcal{P}}^{(i)} \tag{43}$$

for $\hat{\rho}_\Phi^{\text{prob}}$ fixed. Similarly, the training of $\mathcal{L}_{\text{threshold}}$ is to maximize

$$\sum_{i=1}^{H} \hat{\rho}_\Psi^{\text{prob}^{(i)}} \log \hat{\rho}_\psi + \sum_{i=1}^{H} (1 - \hat{\rho}_\Psi^{\text{prob}^{(i)}}) \log(1 - \hat{\rho}_\psi) \tag{44}$$

, where $\hat{\rho}_\Psi^{\text{prob}^{(i)}}$ corresponds to $\hat{\rho}_\Psi^{\text{prob}}$ at $i$-th iteration for $I_{S(n,\rho)\in\mathcal{P}}^{(i)}$. Hence,

$$\arg\min \mathcal{L}_{\text{threshold}} = \frac{1}{H} \sum_{i=1}^{H} \hat{\rho}_\Psi^{\text{prob}^{(i)}} \tag{45}$$

At the global minima of $\mathcal{L}_{\text{threshold}}$, and $\mathcal{L}_{\text{prob}}$,

$$\hat{\rho}_\Psi^{\text{prob}} = \frac{1}{H} \sum_{i=1}^{H} I_{S(n,\rho)\in\mathcal{P}}^{(i)} \tag{46}$$

and

$$\hat{\rho}_\psi = \frac{1}{H} \sum_{i=1}^{H} \hat{\rho}_\Psi^{\text{prob}^{(i)}} \approx \frac{1}{H} \sum_{i=1}^{H} I_{S(n,\rho)\in\mathcal{P}}^{(i)}, \tag{47}$$

for large enough $H$. Let $\xi = \mathbb{P}(S(n,\hat{\rho}_\psi) \in \mathcal{P}) = \frac{1}{H} \sum_{i=1}^{H} I_{S(n,\rho)\in\mathcal{P}}^{(i)} = \arg\min \mathcal{L}_{\text{prob}}$. Choose $p_0 = \xi$. □

**Corollary A.13.** *Fix $\kappa^*, \hat{\rho}_\psi$ and $\hat{\rho}_\Psi^{prob}$. If $\hat{\rho}_\phi \sim Uniform(0,1)$. then the optimal $\hat{\rho}_\phi$ in $\mathcal{L}_{threshold}$ is $q_{0,\kappa^*}$.*

We relax the result of Theorem A.11 for large enough $n$.
*Remark* A.14. $q_{0,\kappa} \lesssim \rho \lesssim p_0$ implies $\mathbb{P}(S(n,\rho) \in \mathcal{F}_\kappa) \geq 1 - \epsilon$

*Proof of Theorem 3.1.* First, we prove the following statement: If the variable values in $S(n,\rho) \in \mathcal{F}_{\kappa^*}$ lead to $\Delta$-optimal solution and $\rho_\Delta^* = \arg\min_\rho \mathbf{c}^\top \mathbf{x}(\tau,\rho)$, then $q_{0,\kappa^*} \lesssim \rho_\Delta^* \lesssim p_0$. To obtain a contradiction, suppose $\rho_\Delta^* = \max(\arg\min_\rho \mathbf{c}^\top \mathbf{x}(\tau,\rho))$. Suppose also $\rho_\Delta^* \lesssim q_{0,\kappa^*}$ or $\rho_\Delta^* \gtrsim p_0$.

1) $\rho_\Delta^* \gtrsim p_0$ implies $S(n,\rho_\Delta^*) \notin \mathcal{P}$ with high probability. This contradicts $\rho_\Delta^* = \max(\arg\min_\rho \mathbf{c}^\top \mathbf{x}(\tau,\rho))$.

2) $\rho_\Delta^* \lesssim q_{0,\kappa^*}$ implies $S(n,\rho_\Delta^*) \notin \mathcal{Q}_{\kappa^*}$ with high probability. Since $q_{0,\kappa^*} \lesssim p_0$, $S(n,\rho_\Delta^*) \in \mathcal{P}$ with high probability. There exists $\rho'$, such that $S(n,\rho') \in \mathcal{Q}_{\kappa^*}$ and $\mathbf{c}^\top \mathbf{x}(\tau,\rho') \leq \mathbf{c}^\top \mathbf{x}(\tau,\rho_\Delta^*)$, and $\rho' > \rho_\Delta^*$. This contradicts $\rho_\Delta^* = \max(\arg\min_\rho \mathbf{c}^\top \mathbf{x}(\tau,\rho))$.

By Proposition A.12,

$$\arg\min_{\hat{\rho}_\psi} \mathcal{L}_{\text{threshold}} = p_0, \text{ for } \hat{\rho}_\phi \text{ and } \hat{\rho}_\Phi^{\text{prob}} \text{ fixed.} \tag{48}$$

By Corollary A.13,

$$\arg\min_{\hat{\rho}_\phi} \mathcal{L}_{\text{threshold}} = q_{0,\kappa^*}, \text{ for } \kappa^*, \hat{\rho}_\psi \text{ and } \hat{\rho}_\Psi^{\text{prob}} \text{ fixed.} \tag{49}$$

By Remark A.14 for large enough $n$,

$$q_{0,\kappa^*} \lesssim \rho \lesssim p_0 \text{ implies } \mathbb{P}(S(n,\rho) \in \mathcal{F}_{\kappa^*}) \geq 1 - \epsilon \tag{50}$$

Thus, $S(n,\rho_\Delta^*) \in \mathcal{F}_{\kappa^*}$ with high probability, for $\rho_\Delta^* \in [q_{0,\kappa^*}, p_0]$. Therefore, we search $\rho_\Delta^* \in [q_{0,\kappa^*}, p_0]$ at the global optimum of $\mathcal{L}_{\text{threshold}}$, such that the cardinality of the search space becomes $(p_0 - q_{0,\kappa^*})|\mathcal{A}|$. □

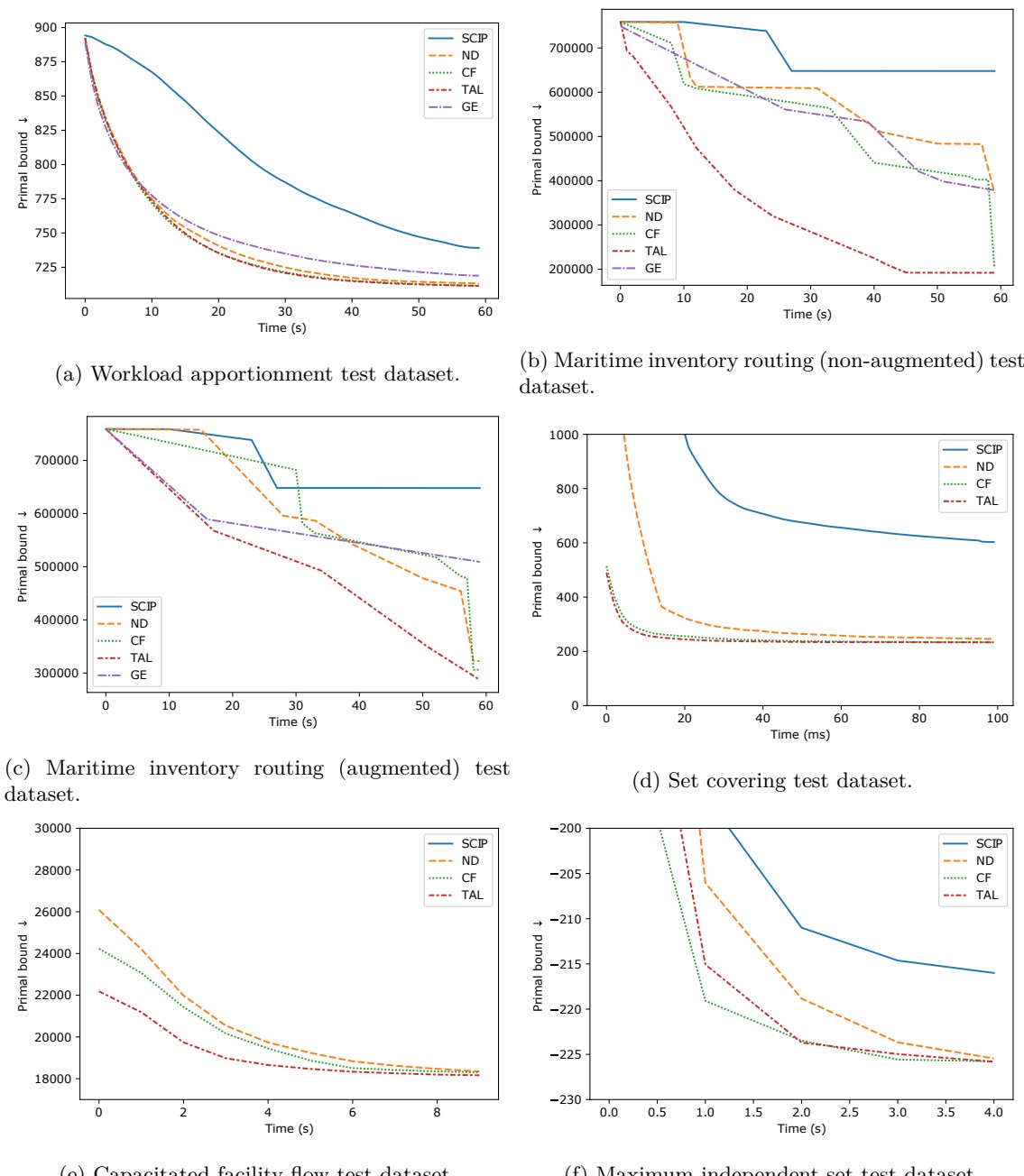

(a) Workload apportionment test dataset.

(b) Maritime inventory routing (non-augmented) test dataset.

(c) Maritime inventory routing (augmented) test dataset.

(d) Set covering test dataset.

(e) Capacitated facility flow test dataset.

(f) Maximum independent set test dataset.

Figure 4: Average primal bound as a function of running time. ND refers to Neural diving (Nair et al., 2020) and GE refers to GNNExplainer (Ying et al., 2019).

## A.5 Average Primal Bound over Time

Figure 4 shows the average primal bound curves for SCIP, ND, GNNExplainer, CF, and TaL. TaL shows the best curves in all test datasets except for maximum independent set problems. In Figure 4e, the SCIP curve is not shown as it is beyond the plotting range. In particular, TaL shows a substantially better curve for maritime inventory routing problems in Figure 4c and 4b.

## A.6 Algorithm Details

---

**Algorithm 2** ThresholdSolve

---

    **Input:**   $M, \mathbf{x}, \hat{\rho}_\psi, \hat{\rho}_\phi, \hat{\rho}_\pi$

1: $\bar{\mathbf{x}} \leftarrow \text{SolveLP}(M, \mathbf{x}, 0)$

2: $\hat{\mathbf{x}}(\hat{\rho}_\pi) \leftarrow \text{SolveLP}(M, \mathbf{x}, \hat{\rho}_\pi)$

3: $\mathbf{x}(\tau, \hat{\rho}_\pi) \leftarrow \text{SolveMIP}(M, \mathbf{x}, \hat{\rho}_\pi, \tau)$

4: $I^{\text{feas}}_{S(n,\hat{\rho}_\psi)} \leftarrow 1$ if $S(n, \hat{\rho}_\psi)$ is feasible else $0$

5: $\mathbf{x}(\tau, \hat{\rho}_\psi) \leftarrow \text{SolveMIP}(M, \mathbf{x}, \hat{\rho}_\psi, \tau)$ if $S(n, \hat{\rho}_\psi)$ is feasible else *Null*

6: $\mathbf{x}(\tau, \hat{\rho}_\phi) \leftarrow \text{SolveMIP}(M, \mathbf{x}, \hat{\rho}_\phi, \tau)$ if $S(n, \hat{\rho}_\phi)$ is feasible else *Null*

7: $\kappa \leftarrow \hat{\rho}_\phi \cdot (\mathbf{c}^\top \mathbf{x}(\tau, \hat{\rho}_\phi) - \mathbf{c}^\top \bar{\mathbf{x}}) + \mathbf{c}^\top \bar{\mathbf{x}}$

8: $I^{\text{LP-sat}}_{S(n,\hat{\rho}_\pi)} \leftarrow 1$ if $\mathbf{c}^\top \hat{\mathbf{x}}(\hat{\rho}_\pi) \geq \kappa$ else $0$

9: $\rho^* \leftarrow \arg\min\limits_{\rho \in [\hat{\rho}_\psi, \hat{\rho}_\phi]} \mathbf{c}^\top \mathbf{x}(\tau, \rho)$ if $S(n, \min(\hat{\rho}_\psi, \hat{\rho}_\phi))$ is feasible else *Null*

10: **return** $I^{\text{feas}}_{S(n,\hat{\rho}_\psi)}, I^{\text{LP-sat}}_{S(n,\hat{\rho}_\pi)}, \rho^*$

---

Algorithm 2 describes the ThresholdSolve procedure at line 8 in Algorithm 7. $\text{SolveLP}(M, \mathbf{x}, 0)$ refers to the process of solving LP-relaxation of $M$ to return the objective value of the solution. $\text{SolveLP}(M, \mathbf{x}, \hat{\rho}_\pi)$ is the process of solving LP-relaxation of the sub-MIP of $M$ by fixing variables with coverage $\hat{\rho}_\pi$ and values $\mathbf{x}$. Since the objective term is unbounded, we set $\kappa \in (\mathbf{c}^\top \bar{\mathbf{x}}, \mathbf{c}^\top \mathbf{x}(\tau, \hat{\rho}_\pi))$, and adjust $\kappa$ to $\kappa^*$.

## A.7 Experiment Details

**Training** The base ND GNN models are trained on 4× NVIDIA Tesla V100-32GB GPUs in parallel. The ND with SelectiveNet models are trained on 4× NVIDIA Tesla A100-80GB GPUs in parallel. The TaL GNN models are trained on a single NVIDIA Tesla A100-80GB GPU. The training times for each dataset and method are as follows. Note that the CPU load is higher than the GPU load in TaL. Also, the TaL column in Table 3 refers to the post-training phase only. It is noticeable that the post-training session does not exceed 2 hours for all datasets.

Table 3: Training times over the methods for each dataset

|  | ND base | ND w/ SelectiveNet | TaL |
|---|---|---|---|
| Workload Apportionment | $\approx 5$ mins | $\approx 3$ hrs | $\approx 1$ hr 10 mins |
| Maritime Inventory Routing (non-augmented) | $\approx 30$ mins | $\approx 2$ hrs 40 mins | $\approx 10$ mins |
| Maritime Inventory Routing (augmented) | $\approx 50$ hrs | $\approx 25$ mins | $\approx 1$ hr 10 mins |
| Set Covering | $\approx 6$ hrs | $\approx 2$ hrs 10 mins | $\approx 1$ hr |
| Capacitated Facility Flow | $\approx 5$ hrs 30 mins | $\approx 12$ hrs | $\approx 5$ mins |
| Maximum Independent Set | $\approx 4$ hrs 50 mins | $\approx 1$ hr | $\approx 5$ mins |

**Evaluation** We apply ML-based methods over presolved models from SCIP, potentially with problem transformations. Throughout the experiments, we are using "aggressive heuristic emphasis" mode in SCIP, where it tunes the parameters of SCIP to focus on finding feasible solutions.

**Adjusting CF cutoff value** In Figure 1b, there is a convex-like region between the confidence score 0.4 and 0.5. We manually find such region and conduct binary search over the region.

**Dataset** The workload apportionment dataset (Gasse et al., 2022) involves problems for distributing workloads, such as data streams, among the least number of workers possible, such as servers, while also ensuring that the distribution can withstand any worker's potential failure. Each specific problem is represented as a MILP and employs a formulation of bin-packing with apportionment. The dataset comprises 10,000 training instances, which have been separated into 9,900 training examples and 100 validation examples.

The non-augmented maritime inventory routing dataset (Gasse et al., 2022; Papageorgiou et al., 2014) consists of problems that are crucial in worldwide bulk shipping. The dataset consists of 118 instances for training, which were split into 98 for training and 20 for validation. Augmented maritime inventory routing dataset includes additional 599 easy to medium level problems from MIPLIB 2017 (Gleixner et al., 2021) on top of the non-augmented maritime inventory routing training set.

The set covering instances (Balas & Ho, 1980; Gasse et al., 2019) are formulated as Balas & Ho (1980) and includes 1,000 columns. The set covering training and testing instances contain 500 rows. The capacitated facility location problems (Cornuéjols et al., 1991; Gasse et al., 2019) are generated as Cornuéjols et al. (1991), and there are 100 facilities (columns). The capacitated facility location training and testing instances contain 100 customers (rows). The maximum independent set problems (Bergman et al., 2016; Gasse et al., 2019) are created on Erdos-Rényi random graphs, formulated as Bergman et al. (2016), with an affinity set at 4. The maximum independent set training and testing instances are composed of graphs with 500 nodes.

