# OpenReview forum: "Threshold-aware Learning to Generate Feasible Solutions for Mixed Integer Programs"
_TMLR — Rejected by TMLR_

### Review · Reviewer_5Hhw · 2023-09-15

**Summary Of Contributions:**

The paper considers the problem of solving the combinatorial optimization problem using neural networks. Specifically, it takes a different approach than the seminal work of Nair et al. (2020) by replacing the SelectiveNet part with a simple thresholding rule. And the threshold value is chosen adaptively based on the underlying problem instance and is modeled by a graph-NN with the input being the problem instance. The paper provides numerical evidence to demonstrate the advantage of the proposed approach.

**Audience:**

Yes

**Broader Impact Concerns:**

NA.

**Claims And Evidence:**

No

**Requested Changes:**

See the weakness in above.

**Strengths And Weaknesses:**

Strengths:

The coverage ratio is indeed an important factor in solving CO with NN. The results in Figure 1 and Figure 2 are interesting and can be inspiring for the algorithm design in practice.

Weakness:

- The paper is hard to follow to some extent. For example,

(i) For key network models (8) and (9), I don't see any description of their architecture. These seem to be different networks than the network for neural diving (ND) as seen from Algorithm 1.

(ii) The definition of (7) is very confusing. As I understand, the probability for predicting the i-th decision variable is given by the ND model $p_{\theta}$, which means there is a different probability of predicting each decision variable. Then why here do all the indices have the same probability $\rho$?

- The positioning of the work is a bit niche. The paper focuses on one single component of the ML4CO which is the coverage ratio when converting the output of the NN model to a candidate warm-start solution for the downstream solver. Arguably, SelectiveNet might not be the best way to do this task. Yet I have two concerns here.

(i) It doesn't seem to me too computationally costly to exhaust all possible threshold values (searching on a fine grid if not all). It costs a linear complexity but it is parallable. For different problem instances, the optimal threshold may vary. So the proposed method, though, has a reduction of this linear complexity, and it indeed suffers from a risk, say, the optimal threshold doesn't lie in the predicted interval.

(ii) The probability output from the ND model (in formula (6)) should be viewed as a score rather than a real probability, because it is not calibrated. This means, for example, if we change the temperature in the training of the generative ND model, all the probabilities will change. In this light, it seems to me not quite stable to provide a "threshold" to cut off the probabilities.

- The theoretical results seem a bit incoherent with the proposed algorithm.

- For the numerical experiments, can you reproduce the experiment setup of Nair et al. (2020).

(i) The current in Section 5.3 seems more like a snapshot. Say, by the time t=10 minutes, reporting the solution quality for all the methods. However, it would be interesting to see how metrics like primal bound change over time such as Nair et al. (2020). Because it may happen that when t=10 minutes, certain method is better, but later on, when t = 20 minutes, another method performs better.

(ii) Is the proposed method compatible with neural branching? In Nair et al. (2020), the combination of neural diving and neural branching gives the best performance for some tasks. It would be a bit unfair to single out the neural diving for a comparison.

(iii) For Figure 4, why do all the primal bounds look so big? Would it still be like this if you continue the x-axis with a longer computation time? Also, why does Figure 4 (f) have negative values on y-axis, should the bound be always positive?

---

> ### Author Response · Authors · 2023-11-20
> **Answer to Reviewer 5Hhw**
>
> Thank you for your valuable input. Our response to each comment is as follows.
> > 1. Specifically, it takes a different approach than the seminal work of Nair et al. (2020) by replacing the SelectiveNet part with a simple thresholding rule. And the threshold value is chosen adaptively based on the underlying problem instance and is modeled by a graph-NN with the input being the problem instance.
>
> __Answer:__ This paper presents two different methods: Confidence Filter (CF) and Threshold-aware Learning (TaL). CF replaces the SelectiveNet with a simple thresholding rule on the neural network output, while TaL replaces SelectiveNet with a GNN that predicts the variable assignment rates (coverage). We will make it more clear in the introduction.
>
> ---
> > 2. For key network models (8) and (9), I don't see any description of their architecture. These seem to be different networks than the network for neural diving (ND) as seen from Algorithm 1.
>
> __Answer:__ We will provide an graphical illustration for the neural network architecture for (8) and (9).
>
> ---
> > 3. The definition of (7) is very confusing. As I understand, the probability for predicting the i-th decision variable is given by the ND model $p_\theta$, which means there is a different probability of predicting each decision variable. Then why here do all the indices have the same probability $\rho$?
>
> __Answer:__ The ND model $p_\theta$ is not the probability for predicting the decision variable, but the probability for the value of the decision variable to be 1. It is true that $p_\theta$ for each decision variable is different. In (7), $\rho$ is the probability that the element is present in the subset, which corresponds to the probability of assigning each decision variable. Here, we assumed $ \rho$ is uniform across all decision variables to provide an insight from threshold function theory as introduced in Section 3.2. We will add a clarification on this for (7).
>
> ---
> > 4. It doesn't seem to me too computationally costly to exhaust all possible threshold values (searching on a fine grid if not all). It costs a linear complexity but it is parallable. For different problem instances, the optimal threshold may vary. So the proposed method, though, has a reduction of this linear complexity, and it indeed suffers from a risk, say, the optimal threshold doesn't lie in the predicted interval.
>
> __Answer:__ It is true that trying all possible threshold values costs linear complexity, which is parallelizable. However, if the number of the decision variables significantly increases, the parallelization cost becomes non-negligible. Indeed, the optimal threshold may vary for different problems. However, our proposed method learns to predict the coverage given the input problem.
>
> ---
> > 5. The probability output from the ND model (in formula (6)) should be viewed as a score rather than a real probability, because it is not calibrated. This means, for example, if we change the temperature in the training of the generative ND model, all the probabilities will change. In this light, it seems to me not quite stable to provide a "threshold" to cut off the probabilities.
>
> __Answer:__ CF does not cut off the probabilities, but does cut off the confidence score as described in the ‘Confidence Filter’ paragraph.

---

> > ### Author Response · Authors · 2023-11-20
> > **Answer to Reviewer 5Hhw (2)**
> >
> > > 6. The current in Section 5.3 seems more like a snapshot. Say, by the time t=10 minutes, reporting the solution quality for all the methods. However, it would be interesting to see how metrics like primal bound change over time such as Nair et al. (2020). Because it may happen that when t=10 minutes, certain method is better, but later on, when t = 20 minutes, another method performs better.
> >
> > __Answer:__ Primal Integral metric captures the time-integral over the primal bound, which is available in Section 5.3. Also, we have added Figure 4 in the appendix to show how primal bound changes over time.
> >
> > ---
> > > 7. Is the proposed method compatible with neural branching? In Nair et al. (2020), the combination of neural diving and neural branching gives the best performance for some tasks. It would be a bit unfair to single out the neural diving for a comparison.
> >
> > __Answer:__ Our proposed method focuses on the primal gap, rather than the dual gap. Therefore, we believe it is fairer to single out the neural diving for comparing the performance on the primal gap, since neural branching is primarily for improving dual gap.
> >
> > ---
> > > 8. For Figure 4, why do all the primal bounds look so big? Would it still be like this if you continue the x-axis with a longer computation time? Also, why does Figure 4 (f) have negative values on y-axis, should the bound be always positive?
> >
> > __Answer:__ The scale of the primal bounds in Figure 4 is based on the objective of the problems from Gasse el al. (2019, 2022), and it would still be similar if we continue the x-axis with a longer computation time. The reason why Figure 4 (f) have negative values on y-axis is that the solution objective values of the problems are calculated to be negative, which does not necessarily be always positive.
> >
> > ---
> > __References:__
> >
> > [1] Maxime Gasse, Didier Chételat, Nicola Ferroni, Laurent Charlin, and Andrea Lodi. Exact combinatorial optimization with graph convolutional neural networks. Advances in Neural Information Processing Systems, 32, 2019.
> > [2] Maxime Gasse, Quentin Cappart, Jonas Charfreitag, Laurent Charlin, Didier Chételat, Antonia Chmiela, Justin Dumouchelle, Ambros Gleixner, Aleksandr M Kazachkov, Elias Khalil, et al. The machine learning for combinatorial optimization competition (ml4co): Results and insights. arXiv preprint arXiv:2203.02433, 2022.

---

### Review · Reviewer_oP85 · 2023-09-19

**Summary Of Contributions:**

This paper studies a learning-based approach for combinatorial optimization that builds on Neural Diving (ND) by Nair et al., 2020. ND provide the model in eq (5) of p(x|M), showing the distribution of feasible solutions given a MIP instance M and uses SelectiveNet to decide which variables should be predicted and which ones should be delegated to a smaller MIP solver. This paper proposes alternatives to this portion that use confidence filters and threshold aware learning. Algorithm 1 summarizes the main threshold-aware learning and Table 2 summarizes the experimental results measuring the solution quality of the learned solver.

**Audience:**

Yes

**Broader Impact Concerns:**

I have no concerns

**Claims And Evidence:**

No

**Requested Changes:**

For acceptance, I would like a convincing comparison and connection to other published results, and for the usage of the ML4CO dataset to be changed and the SOTA claims to be toned down.

**Strengths And Weaknesses:**

*Strengths.*
1. Dissecting components of larger systems like Neural Diving is a relevant and important topic for learning MIP solvers. Generally improving MIP solvers with ML methods is a very impactful direction.
2. The idea of ablating Neural Diving's SelectiveNet architecture is reasonable

*Weaknesses.*
1. The largest weakness is in the experimental evaluation. Table 2 is not directly comparable to other published work, so evaluating the experimental setup is not easy. My concern is that the [ML4CO competition](https://www.ecole.ai/2021/ml4co-competition/) took place in 2021, and there they had the same evaluation setup that was fairly running and comparing every submitted method on the same hardware. Now that the competition is over, while it would be good for the community to use it as a benchmark, is seems difficult to compare methods on it. Furthermore, this paper does not compare with the winning method of the benchmark (CUHKSZ_ATD). Thus, I believe the claim in the abstract and throughout the paper of attaining state-of-the-art performance on this benchmark is not accurate.
2. The experimental comparisons to Neural Diving is also not easy to evaluate or compare to the settings from the original paper. This is not ideal as ND has some hyper-parameters that need to be tuned (e.g., coverage thresholds) and it is not clear the results presented in this paper are a fair comparison. Portions of the ND code and datasets are available [here](https://github.com/google-deepmind/deepmind-research/tree/master/neural_mip_solving), and I would have found it significantly more convincing to directly compare to all of their publicly available models/datasets.

---

> ### Author Response · Authors · 2023-11-20
> **Answer to Reviewer oP85**
>
> We appreciate your valuable comments. Our response to each comment is as follows.
>
> > 1. The largest weakness is in the experimental evaluation. Table 2 is not directly comparable to other published work, so evaluating the experimental setup is not easy. My concern is that the ML4CO competition took place in 2021, and there they had the same evaluation setup that was fairly running and comparing every submitted method on the same hardware. Now that the competition is over, while it would be good for the community to use it as a benchmark, is seems difficult to compare methods on it. Furthermore, this paper does not compare with the winning method of the benchmark (CUHKSZ_ATD). Thus, I believe the claim in the abstract and throughout the paper of attaining state-of-the-art performance on this benchmark is not accurate.
>
> __Answer:__ We agree that it is difficult to compare methods on the exact same environment as ML4CO competition. Also, it is true that our method does not compare with the winning method of the competition (CUHKSZ_ATD) as technical details of the winning method is not released. Therefore, we will tone down our SOTA claims.
>
> ---
> > 2. The experimental comparisons to Neural Diving is also not easy to evaluate or compare to the settings from the original paper. This is not ideal as ND has some hyper-parameters that need to be tuned (e.g., coverage thresholds) and it is not clear the results presented in this paper are a fair comparison. Portions of the ND code and datasets are available here, and I would have found it significantly more convincing to directly compare to all of their publicly available models/datasets.
>
> __Answer:__ Thank you for your suggestion to compare our method to the dataset used in Nair et al. (2019). However, the technical details of ND training and ND models are not publicly available (e.g. SelectiveNet coverage and other training hyperparameters) in their original paper as well, which makes reproducing the results in Nair et al. (2019) practically challenging.
>
> ---
> __References:__
> [1] Vinod Nair, Sergey Bartunov, Felix Gimeno, Ingrid von Glehn, Pawel Lichocki, Ivan Lobov, Brendan O’Donoghue, Nicolas Sonnerat, Christian Tjandraatmadja, Pengming Wang, et al. Solving mixed integer programs using neural networks. arXiv preprint arXiv:2012.13349, 2020.

---

> > ### Comment · Reviewer_oP85 · 2023-11-20
> >
> > Thank you for the clarifications. I have re-read the paper and gone through the responses and other reviews. Unless there is anything else to discuss, I will maintain my original assessment as the paper still seems to be missing a sound comparison to the related literature and approaches.

---

> > > ### Author Response · Authors · 2023-11-21
> > >
> > > Thank you for your careful reading. Despite the practical challenges in reproducibility, we will try to compare our methods against ND in the neural network verification dataset within the discussion period.

---

> ### Author Response · Authors · 2023-12-02
> **Answer to Reviewer oP85**
>
> The result comparing our methods to ND on the neural net verification dataset (200 test instances) within the two-minute time limit is as follows:
> |                                 | PI ↓       | PB ↑        |
> |---------------------------------|------------|-------------|
> | SCIP                            | 2088.44    | **-13.042** |
> | Neural diving (Nair et al.)     | 1910.22    | -13.62      |
> | Confidence Filter (Ours)        | **1906.2** | -13.68      |
> | Threshold-aware Learning (Ours) | 1945.981   | -13.62      |
>
> Please note that the objective sense is to maximize on this dataset, i.e., the solution with a higher primal bound is considered the better solution. Here, CF outperformed ND in primal integral, while TaL did not. We suppose that the result of ND outperforming TaL in this comparison is due to the large variance over the optimal coverage of the model trained on this dataset. We will include this result and TaL's limitation in the paper.

---

### Review · Reviewer_nqJA · 2023-11-06

**Summary Of Contributions:**

This paper develops a technique to find a high-quality solution to a mixed integer program (MIP) in limited time. It specifically considers approaches where a subset of the decision variables are assigned quickly using a learned model (e.g., Neural Diving), leaving the remainder to be solved for using an off-the-shelf MIP solver. The authors consider two approaches that control the “coverage” of the initial variable assignments to be made by the learned model. A higher coverage implies relying on the learned model more which typically gives a faster, but less accurate solution. The first approach they consider incorporates a confidence filter (CF). CF suggests variables for assignment if the confidence of ND’s assignment is above a given threshold. Alternatively, the authors also propose threshold-aware Learning (TaL). TaL models the probability of assigning a variable using a separately learned graph neural-network model that takes the MIP instance as input. This probability threshold is modelled to be the same for all variables, and all variables are selected for assignment independently. The authors show the choice of target confidence score and probability threshold is important for the task and admits a “sweet spot”. CF performs decently well on average, however, TaL outperforms it along with all prior baselines across 6 tasks.

**Audience:**

Yes

**Claims And Evidence:**

Yes

**Requested Changes:**

1) In the introduction, you refer to “higher variable value classification accuracy” and “correctly predicted variable values”. What is a correctly predicted variable value? I would have thought an individual variable must be judged in combination with the remainder of the variables. Do you assume the solution to the combinatorial optimization problem is unique? Can you please make clear what you mean by this terminology in the introduction?

2) In the introduction, can you introduce briefly what threshold function theory is? Overall, I felt the introduction could benefit from a more high level description of relevant topics before diving into specifics and bespoke terminology.

3) In section 3, the confidence score lies in [0.5, 1.0], no? Why then do we see values in [0, 0.5] on the x-axis of Figure 1? What are we seeing in this Figure? Is the x-axis not $\Gamma$? Please include corresponding greek symbol labels with the axis descriptions to make this clearer. E.g., is the x-axis in Figure 2 reporting $\rho$?

4) Coverage is defined at the end of section 2. Given that this concept is central to the paper and that it doesn’t rely on much more than a shallow understanding of combinatorial optimization and variable assignment, I would advise moving this up to early in the introduction.

**Strengths And Weaknesses:**

I found the proposed approach intuitive, yet also surprisingly effective given its simplicity. While the TaL approach relies on training a GNN which is "complex", I would not have predicted that an independent probability of variable assignment would be sufficient for strong performance.

In terms of weaknesses, I think the introduction could benefit from additional background and explanation. I go into more detail in requested changes below. I was also confused by a some of the ranges of variables.

---

> ### Author Response · Authors · 2023-11-20
> **Answer to Reviewer nqJA**
>
> We would like to express our gratitude to your review. Our response to each comment is as follows.
>
> > 1. In the introduction, you refer to “higher variable value classification accuracy” and “correctly predicted variable values”. What is a correctly predicted variable value? I would have thought an individual variable must be judged in combination with the remainder of the variables. Do you assume the solution to the combinatorial optimization problem is unique? Can you please make clear what you mean by this terminology in the introduction?
>
> __Answer:__ By “higher variable value classification accuracy” and “correctly predicted variable values”, we intend to refer to the collected solution data for training the Neural Diving model as the reference solution. In the introduction, we will fix these expressions more clearly.
>
> ---
> > 2. In the introduction, can you introduce briefly what threshold function theory is? Overall, I felt the introduction could benefit from a more high level description of relevant topics before diving into specifics and bespoke terminology.
>
> __Answer:__ In the introduction, we will add the explanations on what threshold function theory is.
>
> ---
> > 3. In section 3, the confidence score lies in [0.5, 1.0], no? Why then do we see values in [0, 0.5] on the x-axis of Figure 1? What are we seeing in this Figure? Is the x-axis not $\Gamma$? Please include corresponding greek symbol labels with the axis descriptions to make this clearer. E.g., is the x-axis in Figure 2 reporting $\rho$?
>
> __Answer:__ It is true that the confidence score lies in [0.5, 1]. We will fix the x-axis range of Figure 1 to [0.5, 1] and include $\Gamma$ on the x-axis of Figure 1. Yes, the x-axis in Figure 2 reports $\rho$, and we will also include $\rho$ on the x-axis of Figure 2.
>
> ---
> > 4. Coverage is defined at the end of section 2. Given that this concept is central to the paper and that it doesn’t rely on much more than a shallow understanding of combinatorial optimization and variable assignment, I would advise moving this up to early in the introduction.
>
> __Answer:__ Thank you for your suggestion. we will move the definition of coverage to early in the introduction.

---

> > ### Comment · Reviewer_nqJA · 2023-12-06
> >
> > Thank you for your explanations and amendments. I maintain my score.

---

### Decision · Action_Editor_PZEF · 2023-12-20

**Recommendation:** Reject

**Comment:**

All reviewers agreed that the direction of the work was worthwhile and that the methods themselves were of thought-provoking, but, after the rebuttal, multiple reviewers had continuing concerns about the presentation, evidence, and claims. These concerns were too substantial to accept the work as is or with minor revisions.

Updates that would be expected in a resubmission with major revision:
1. Completing the comparison to prior work that was started during the rebuttal period. Integrating learnings from these experiments into the paper, including a discussion of possible limitations that were brought to light. For example, in the rebuttal, the author's discuss a large variance in optimal coverage that could lead to unexpected results, but there is no discussion of variance in the paper; adding some form of confidence bounds (due to model variance) to the numerical tables in the paper could go a long way in helping readers accept the significance of the results.
2. Completing the suggested reorganization of the early part of the paper to make the presentation/motivation/goals more clear. This includes adding details like the particular neural architectures used.
3. Modifying the claims throughout the paper based on the experimental results and specific paper goals. For example, instead of emphasizing claims of state-of-the-art, centering the claims around the surprising fact that a simplification of ND can lead to better primal bounds and performance as good or better than ND in a specific set of scenarios.

In the authors' rebuttal, they provided helpful answers on many of the points raised by reviewers, and said at a high level how they planned to revise the paper. However, no updated version of the paper has been uploaded, and the revisions will require substantial updates that should be reviewed.

**Audience:**

All reviewers agreed that the direction of the work was worthwhile and of interest to others in the TMLR community.

**Claims And Evidence:**

As is, the claims are too strong or unclear given the presented evidence. To address this concern, more work would need to be done to clarify the presentation/motivation/goals of the approach, include clearer comparison to the baseline approach to support the paper's hypothesis, and modify the claims throughout to reflect the evidence.

**Resubmission Of Major Revision:**

The authors may consider submitting a major revision at a later time.